

# Multi-scale dynamics of carbon dioxide flux and its environmental drivers in the Pantanal wetland

Tarcis A. O. dos Santos[1,2], Alberto S. de Arruda[2], Paulo H. Z. de Arruda[2], and Gilney F. Zebende[2,3]

[1]State University of Mato Grosso, Ingás Av., Sinop, 78555, MT, Brazil
[2]Federal University of Mato Grosso, Fernando Corrêa Av., Cuiabá, 78060, MT, Brazil
[3]State University of Feira de Santana, Transnordestina Av., Feira de Santana, 44036, BA, Brazil

**Correspondence:** Tarcis A. O. dos Santos (tarcis@unemat.br)

**Abstract.** Understanding carbon flux dynamics in tropical ecosystems is crucial for evaluating their role in global climate regulation. This study investigates the temporal variability of the net ecosystem exchange ($NEE$) of $CO_2$ and its interactions with key meteorological variables in a tropical forest ecosystem of the Pantanal, Brazil. Using high-resolution hourly data from a flux tower, we apply Detrended Fluctuation Analysis (DFA) and Detrended Cross-Correlation Analysis (DCCA) to analyze

diurnal to seasonal cycles of $NEE$, latent heat ($LE$), sensible heat ($H$), global radiation ($Rg$), air and soil temperature ($T_{air}$ and $T_{soil}$), relative humidity ($rH$), and vapor pressure deficit ($VPD$). The results reveal a strong diurnal coupling between solar radiation, temperature, and carbon fluxes, with peak $CO_2$ uptake occurring at midday. A key novel finding is a marked shift to strong anti-persistence in $NEE$ at the weekly scale during the dry season, a pattern supported by concurrent reductions in $LE$ and $rH$ and increases in $H$ and $VPD$. This highlights that water limitation is a critical driver of carbon release and reveals

a previously unidentified regulatory mechanism in the ecosystem's carbon cycle. These findings underscore the sensitivity of carbon dynamics to hydrometeorological conditions and underline the necessity of multi-scale analysis. Uncertainties remain regarding the role of extreme droughts and floods, as well as land-use dynamics, which merit further investigation.

## 1 Introduction

The Pantanal is one of the largest floodplains in the world, located in the center of South America, covering approximately

$160\,000\ \mathrm{km}^2$ across Brazil, Bolivia, and Paraguay (Teodoro et al., 2016). About 40% of this territory lies within Brazil, encompassing the states of Mato Grosso and Mato Grosso do Sul (da Silva and de Moura Abdon, 1998). It is a significant sedimentary basin whose ecological dynamics are deeply influenced by climatic variables such as precipitation, temperature, and humidity, which shape the seasonal flood regimes that sustain its highly diverse flora and fauna (Louzada et al., 2020).

Although the Pantanal holds global ecological and climatic importance, there is still a lack of systematic studies using high-

resolution quantitative data that allow robust modeling of climate–ecosystem interactions (Teodoro et al., 2016; Pobocikova et al., 2021). Understanding these interactions is essential, especially in the context of climate change, given the role that tropical ecosystems play in climate regulation—through $CO_2$ uptake via photosynthesis, emission by respiration and decomposition, as well as processes like evapotranspiration, albedo, and heat flux. These processes directly affect the global carbon balance and the Earth's climate system.



In this study, we aim to characterize the temporal variability and interdependence between the Net Ecosystem Exchange of carbon ($NEE$) and meteorological variables in a representative area of the Pantanal in Mato Grosso. The data were obtained using the Eddy Covariance technique, which enables direct measurement of $CO_2$ exchanges between the ecosystem and the atmosphere at high temporal resolution. This allows for multiscale analysis (hourly, weekly, monthly, and seasonal), which is essential to detect patterns of persistence, trend, and correlation among variables.

In this research, we applied statistical methods suited to the non-stationary nature of climatic and ecological time series, such as Detrended Fluctuation Analysis (DFA) (Peng et al., 1994) to investigate long-range autocorrelation in individual series, and Detrended Cross-Correlation Analysis (DCCA) (Podobnik and Stanley, 2008), whose extension enables the assessment of the strength of the relationship between series pairs using the $\rho_{\mathrm{DCCA}}$ coefficient (Zebende, 2011). The results contribute to a more refined understanding of the Pantanal ecosystem's response to microclimatic variability, providing scientifically relevant insights for conservation strategies, monitoring efforts, and mitigation of climate change effects in the region.

## 2    Materials and Methodology

### 2.1    Materials

This study aimed to investigate the interdependence between the Net Ecosystem Exchange of carbon ($NEE$) and relevant environmental variables, listed in Table 1, including air temperature, solar radiation, relative humidity, sensible heat, latent heat, and vapor pressure deficit. The data were collected using the *Eddy Covariance* technique, through a micrometeorological tower installed in a seasonally flooded forest (*mata galaria*) at the Baía das Pedras site ($16°29'53''S$; $56°24'46''W$), located in the northern Pantanal, 130 km southwest of Cuiabá, Mato Grosso, Brazil.

The tower, equipped with high-precision sensors, was configured with an infrared gas analyzer (LI-7500) for continuous measurements of $CO_2$ and water vapor, as well as three-dimensional sonic anemometers (WindMaster, R.M. Young; CSAT3, Campbell Scientific) to record the wind components. The sensors were mounted at a height of 20 meters, and data were recorded at a frequency of 10 Hz. The initial processing of raw data was performed using EddyPro software, with corrections applied for air density fluctuations, spectral losses, frequency response, and temperature. Quality filters were used to remove noise, instrumental failures, non-stationary data, and values outside plausible ranges. After these steps, approximately 70% of the data were retained, with gaps filled using the marginal distribution method, as described by (Dalmagro et al., 2022).

Fig. 1 presents the complete time series of the variables investigated. The combination of the *Eddy Covariance* technique with robust statistical methods enabled the construction of a reliable and high-resolution dataset, suitable for analyzing climate patterns, seasonal variations, and interactions between energy and mass fluxes in the ecosystem.

To investigate the interdependencies among variables, we applied three complementary approaches: Detrended Fluctuation Analysis (DFA), Detrended Cross-Correlation Analysis (DCCA), and the $\rho_{\mathrm{DCCA}}$ coefficient. DFA allows the identification of long-term auto-correlation within a single time series, while DCCA detects the presence of cross-correlations between two time series. The $\rho_{\mathrm{DCCA}}$ coefficient quantifies the level of these cross-correlations, providing a normalized metric ranging from





**Table 1.** Variables under study

| Range of $\alpha$ | Interpretation |
| --- | --- |
| $\alpha = 0.5$ | White Noise: No correlation, random values. |
| $0 < \alpha < 0.5$ | Anti-persistence: High values are followed by low values and vice versa. |
| $0.5 < \alpha < 1$ | Persistence: High values are followed by high values and vice versa. |
| $\alpha \approx 1$ | 1/f Noise: Long-range correlations. |
| $\alpha > 1$ | Non-stationary: Trend present, variance increases over time. |

The eight environmental variables collected in the Pantanal and the meaning of each symbol.

$-1$ to $1$. The joint application of these methods offers a deeper understanding of the dynamics that regulate $CO_2$ fluxes and their relationship with microclimatic factors in the Pantanal.

## 2.2 Methodology

To investigate temporal correlations and interdependencies among the variables, two main statistical methods and a coefficient were employed: Detrended Fluctuation Analysis (DFA), Detrended Cross-Correlation Analysis (DCCA), and the cross-correlation coefficient $\rho_{\text{DCCA}}$. These methods are suitable for handling non-stationary time series, allowing the detection of persistence patterns and correlations across different temporal scales.

### 2.2.1 DFA (Detrended Fluctuation Analysis)

Since the pioneering work of Hurst (1951), the analysis of long-range correlations (LRC) in time series has become a fundamental tool for characterizing temporal dependence in complex systems. The Hurst exponent, $h$, quantifies the degree of persistence or anti-persistence of fluctuations over time and is directly associated with the decay of auto-correlations as the time lag increases. Originally proposed in hydro-logical studies of the Nile River, the exponent has since been widely used to detect long-term memory in various classes of natural and socioeconomic systems.

Values of $h$ in the range $0.5 < h < 1.0$ indicate persistence—fluctuations in the same direction tend to cluster—while values $h < 0.5$ indicate anti-persistence, characterized by frequent reversals in fluctuation direction. The case $h = 0.5$ corresponds to white noise, typical of short-memory processes in which auto-correlations decay exponentially. Beyond its relationship with statistical memory, the Hurst exponent is also linked to the fractal dimension of the series and is widely used in fields such as hydrology (Koutsoyiannis, 2003), finance (Couillard and Davison, 2005; Bassler et al., 2006), and nonlinear systems analysis

(Matcharashvili and Prangishvili, 2020).

To quantify LRC in time series that exhibit trends or non-constant fluctuations — that is, non-stationary series — a widely recognized approach is Detrended Fluctuation Analysis (DFA), proposed by Peng et al. (1994). This method removes local trends over various time windows and computes the average residual fluctuation as a function of scale, enabling a reliable estimation of the Hurst exponent even in the presence of non-stationarity.



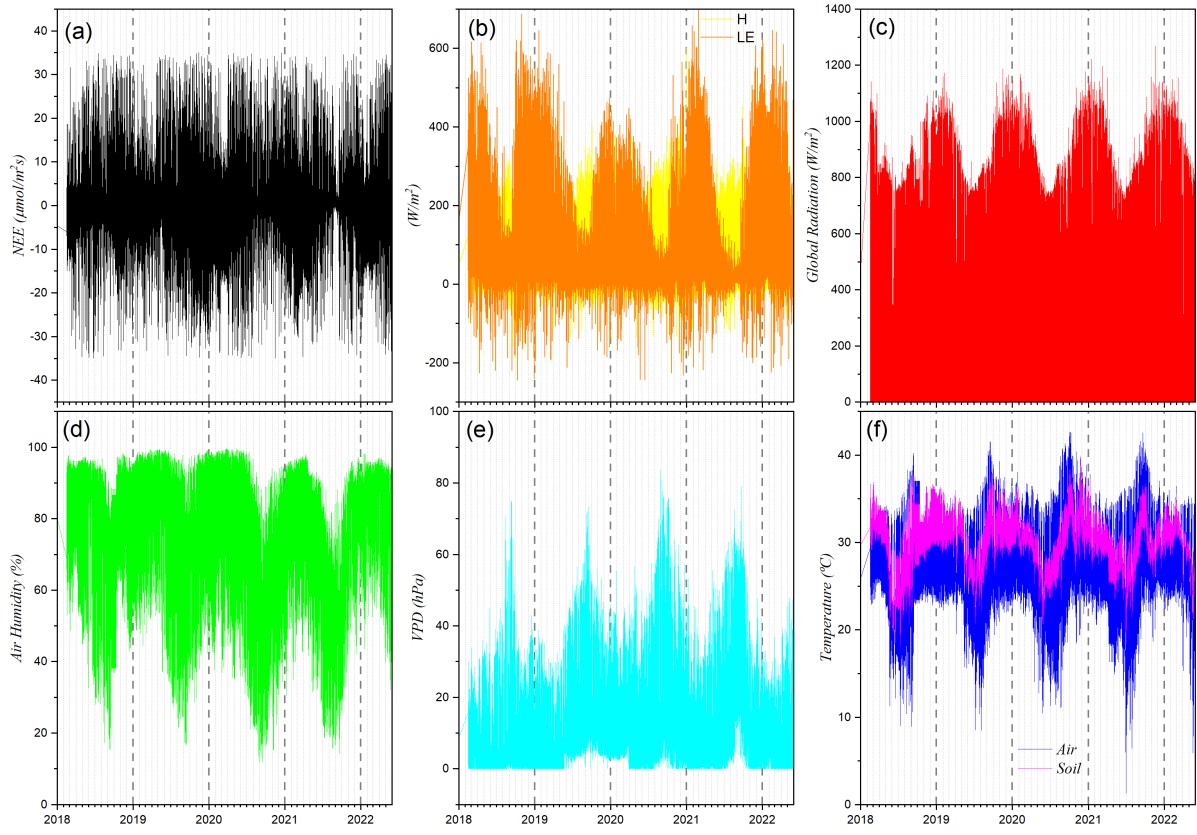

**Figure 1.** Time series of (a) Net Ecosystem Exchange ($NEE$), (b) sensible ($H$) and latent ($LE$) heat fluxes, (c) global radiation, (d) relative air humidity, (e) vapor pressure deficit ($VPD$), and (f) air and soil temperature for the Baía das Pedras site (Northern Pantanal) as a function of time.

The importance and robustness of DFA have been widely acknowledged in the scientific literature, with applications in physiological, climatic, ecological, and financial data. The method quantifies long-term auto-correlation in a single time series $\{x_k\}$, for $k = 1, \ldots, N$, providing a robust measure of the persistence level in the series' variations. The interpretation of the obtained $h$ values can be found in Table 2, as suggested by (Hu et al., 2001; Kantelhardt et al., 2001).

To precisely quantify this level of persistence, the DFA algorithm systematically processes the time series. The core procedure for calculating the scaling exponent $\alpha$ is detailed as follows:





1. **Profile Creation (Integration):** Starting from the mean value, $\langle x \rangle = \frac{1}{N}\sum_{k=1}^{N} x_k$, the series is demeaned and the result is cumulatively summed, generating a new series $X(i)$, called the integrated time series:

$$X(i) = \sum_{k=1}^{i}[x_k - \langle x \rangle], \quad \text{for } i = 1, \ldots, N \tag{1}$$

2. **Segmentation:** The profile $X(i)$ is divided into $(N-n)$ overlapping segments of equal size $n$ (temporal scale). In this case, each segment $k$ contains $(n+1)$ values.

3. **Detrending:** In each segment, a local trend is removed from the data by fitting a polynomial of order $m$, resulting in the adjusted ordinate value $\widetilde{X}(i,k)$ using the least squares method.

4. **Calculation of Local Fluctuation:** For each segment $k$, the variance of the residuals is calculated as:

$$f_{DFA}^2(n,k) = \frac{1}{n+1}\sum_{j=i}^{i+n}[X(j) - \widetilde{X}(j,k)]^2 \tag{2}$$

5. **Calculation of Mean Fluctuation:** The fluctuation function for each time scale $n$ is given by:

$$F_{DFA}(n) = \sqrt{\frac{1}{(N-n)}\sum_{k=1}^{(N-n)} f_{DFA}^2(n,k)} \tag{3}$$

This process is repeated for other temporal scales, with $4 \leq n \leq N/4$.

6. **Power-Law Analysis:** If the time series exhibits long-term correlations, $F_{DFA}$ will follow a power-law function of $n$. In a log-log plot of $F(n)$ versus $n$, this may be represented by a linear relationship, that is:

$$F_{DFA} \propto n^\alpha \tag{4}$$

Here, $\alpha$ is the autocorrelation exponent and the main result of the DFA method, as described in Table 2, with a good introduction in (Løvsletten, 2017).

### 2.2.2 DCCA Method and Coefficient $\rho_{\text{DCCA}}$

DCCA is a generalization of the DFA method for analyzing cross-correlations between two time series, $\{x_k\}$ and $\{y_k\}$, of equal length $N$, at different time scales $n$, with removal of local trends. The method is described in detail in (Podobnik and Stanley, 2008), and comprises the following algorithm.

1. **Profile Creation (Integration):** As in DFA, the two series are integrated separately, that is, based on the mean values, $\langle x \rangle = \frac{1}{N}\sum_{k=1}^{N} x_k$ and $\langle y \rangle = \frac{1}{N}\sum_{k=1}^{N} y_k$, the series are demeaned and then accumulated, resulting in two new time series, $X(i)$ and $Y(i)$, called integrated series:

$$X(i) = \sum_{k=1}^{i}[x_k - \langle x \rangle] \quad \text{and} \quad Y(i) = \sum_{k=1}^{i}[y_k - \langle y \rangle] \quad \text{for } i = 1, \ldots, N \tag{5}$$





**Table 2.** Interpretation of $\alpha$ exponent from DFA method.

| Value of $\rho_{\text{DCCA}}$ | Interpretation |
| --- | --- |
| -1.000 | perfect anti cross-correlation |
| (-1.000; -0.666] | strong anti cross-correlation |
| (-0.666; -0.333] | moderate anti cross-correlation |
| (-0.333; 0.000) | weak anti cross-correlation |
| 0.000 | no cross-correlation |
| (0.000; 0.333] | weak cross-correlation |
| (0.333; 0.666] | moderate cross-correlation |
| (0.666; 1.000) | strong cross-correlation |
| 1.000 | perfect cross-correlation |

2. **Segmentation and Detrending:** Steps 2 and 3 of DFA are independently applied to both profiles, $X(i)$ and $Y(i)$.

3. **Fluctuation Covariance Calculation:** Instead of variance, DCCA calculates the covariance between the residuals of the two series in each segment $k$:

$$f_{DCCA}^2(n,k) = \frac{1}{n+1} \sum_{j=i}^{i+n} [X(j) - \widetilde{X}(j,k)][Y(j) - \widetilde{Y}(j,k)] \qquad (6)$$

4. **Average Covariance:** The average covariance for scale $n$ is obtained by averaging over all segments:

$$F_{DCCA}^2(n) = \frac{1}{(N-n)} \sum_{k=1}^{(N-n)} f_{DCCA}^2(n,k) \qquad (7)$$

5. **Calculation of the Coefficient $\rho_{\text{DCCA}}$ (Zebende, 2011):** The DCCA cross-correlation coefficient is calculated using the equation:

$$\rho_{DCCA}(n) = \frac{F_{DCCA}^2(n)}{F_{DFA_x}(n) F_{DFA_y}(n)} \qquad (8)$$

Where $F_{DFA_x}(n)$ and $F_{DFA_y}(n)$ are the DFA fluctuations calculated for the series $\{x_k\}$ and $\{y_k\}$ respectively.

The interpretation of possible values of $\rho_{\text{DCCA}}$ is shown in Table 3, noting that $\rho_{DCCA}(n)$ ranges from -1 to 1. This was first postulated in Zebende (2011), in face of others conventional metrics. The main advantage of using the coefficient $\rho_{\text{DCCA}}$ lies in its ability to quantify the level of cross-correlation between non-stationary time series, where traditional correlation measures (e.g., Pearson's coefficient) fail due to their sensitivity to trends and nonstationarity. Unlike the classical cross-correlation function, which assumes stationarity, $\rho_{\text{DCCA}}$ is scale-dependent and normalized, ranging from $-1$ to 1, making it suitable for detecting and quantifying correlations embedded in power-law noise and non-stationary signals. Furthermore, as shown in Zebende (2011), $\rho_{\text{DCCA}}$ establishes a direct relationship between the long-range auto-correlation exponents $\alpha_1, \alpha_2$ and the



**Table 3.** Interpretation of $\rho_{\mathrm{DCCA}}$ Coefficient Values

| Index | Daily | Weekly | | Monthly | Quarterly |
|-------|-------|--------|------|---------|-----------|
| $NEE$ | 0.96 | 0.24 | | 0.85 | 1.07 |
| $H$ | 1.31 | 0.19 | | 0.67 | 1.23 |
| $Rg$ | 1.40 | 0.17 | | 0.52 | 0.81 |
| $LE$ | 1.33 | 0.25 | | 0.80 | 1.39 |
| $rH$ | 1.48 | 0.46 | | 0.85 | 1.43 |
| $T_{air}$ | 1.51 | 0.32 | 0.79 | 0.71 | 1.23 |
| $T_{soil}$ | 1.55 | 0.42 | 1.05 | 1.02 | 1.45 |
| $VPD$ | 1.43 | 0.40 | | 0.78 | 1.36 |

cross-correlation exponent $\lambda$, allowing a consistent and robust interpretation of long-range interactions that cannot be captured by conventional metrics.

The DCCA cross-correlation coefficient ($\rho_{\mathrm{DCCA}}$) is a robust tool for quantifying the relationship between two non-stationary time series, finding vast application in hydrological and climate studies. For instance, it has been used to quantify the cross-correlation between air temperature and relative humidity in various global locations, showing that their relationship varies significantly and is influenced by seasonal patterns (Vassoler and Zebende, 2012). In a similar approach, cross-correlation has been used in case studies related to water security issues (Fernandez et al., 2024). In the Brazilian context, the $\rho_{\mathrm{DCCA}}$ coefficient

has also served as the fundamental metric for constructing complex climate networks, allowing for an in-depth analysis of the interconnections between different locations based on their climate data (Oliveira Filho et al., 2023).

The applicability of $\rho_{\mathrm{DCCA}}$ extends globally to investigations of climate patterns across different spatial and temporal scales. A study on temperature records in China used DCCA to reveal different spatial cross-correlation patterns across multiple time scales (Yuan and Fu, 2014). Analogously, the technique has been used to uncover the correlation patterns between a large-scale

climate index, such as the North Atlantic Oscillation (NAO), and precipitation, demonstrating its effectiveness in connecting global phenomena with local meteorological variables (Tatli and Menteş, 2019).

The DCCA cross-correlation coefficient has been widely employed to investigate contagion and interdependence in economic systems. Applications range from the Brazilian stock market, where correlations between the Ibovespa index and blue-chip stocks strengthened after the 2008 crisis (da Silva et al., 2015), to international studies introducing $\Delta\rho_{\mathrm{DCCA}}$ as a measure

of crisis-driven contagion across G7 countries (da Silva et al., 2016). Beyond financial markets, the method has revealed a scale-dependent negative relationship between oil prices and the US dollar exchange rate (Reboredo et al., 2014). Recent refinements, such as the sliding-window approach, further allow mapping the temporal evolution of correlations, enhancing the capacity of DCCA to capture dynamic interactions in macroeconomic contexts (Guedes and Zebende, 2019).





## 3  Results

### 3.1  Preliminary Results

Before applying advanced statistical methods (DFA, DCCA, and $\rho_{\mathrm{DCCA}}$), a classical descriptive analysis of the variables was conducted. This step is fundamental for characterizing data behavior, validating its quality, and establishing the premises for subsequent analyses.

#### 3.1.1  Descriptive Statistics of the Time Series

Based on observations collected by the micrometeorological station, time series were constructed for the eight analyzed variables, as illustrated in Fig. 1. Among them, we highlight the Net Ecosystem Exchange (NEE) as the dependent variable. This variable exhibits a marked diurnal pattern: during the day, it tends to negative values, indicating $CO_2$ uptake by vegetation through photosynthesis; at night, values become positive, reflecting $CO_2$ release by respiration. Fig. 1(a) also reveals seasonal fluctuation, with NEE values ranging approximately from $-40$ to $+40$ $\mathrm{umolm}-2\mathrm{s}-1$, consistent with the annual vegetation growth cycles.

Figures 1(b) and 1(c) show the sensible heat flux ($H$), latent heat flux ($LE$), and global radiation ($Rg$), measured in $\mathrm{Wm}-2$. These variables display well-defined diurnal cycles, with peaks during the day and values close to zero at night. $Rg$ exhibits pronounced seasonality, with peaks during summer (December to February) and minimum values in winter (June to August), a pattern also observed for $H$ and $LE$, though with greater variability in the hotter months.

Relative humidity ($rH$), shown in Fig. 1(d), is higher during the night and early morning, approaching 100%, and declines throughout the day as temperature rises. The vapor pressure deficit ($VPD$), shown in Fig. 1(e), displays the opposite pattern, reaching its highest values during the day, reflecting increased evaporative demand of the air. Fig. 1(f) illustrates the diurnal cycles of air and soil temperature: the air warms and cools more quickly, whereas the soil, due to its higher thermal inertia, shows lower amplitude and thermal lag, sometimes surpassing air temperature.

Table 4 summarizes the main indicators of descriptive statistics for the investigated variables. For NEE, a slightly positive mean (0.54) is observed, while the median (1.44) and mode (-10.10) suggest an asymmetric distribution. The other variables follow patterns consistent with regional climatology, exhibiting high values for $LE$ and $Rg$, broad thermal variation, and moderate asymmetries.

Descriptive statistics applied to the atmospheric dataset allow for an objective synthesis of variability and predominant patterns throughout the sampling period. Metrics such as mean, median, mode, standard deviation, skewness, and kurtosis enable the characterization of central tendencies, fluctuation amplitude, and distribution shape, contributing to the identification of anomalies and validation of data quality. This initial characterization provides a solid foundation for subsequent analyses, guiding the selection and interpretation of advanced analytical methods applied in this study to investigate interdependencies and correlations among variables.

The initial descriptive analysis presented in Table 4 characterized the individual distributions of the eight analyzed atmospheric variables based on metrics such as mean, median, mode, standard deviation, skewness, and kurtosis. This step provided




a preliminary understanding of central trends and data variability, identifying relevant patterns and possible distributional asymmetries.

To further characterize these patterns and investigate their variation over the 24-hour cycle, Fig. 2 was developed, showing the hourly means of these variables. This graphical representation is fundamental for identifying characteristic diurnal cycles, such as air heating and cooling processes, relative humidity oscillations, energy flux fluctuations, and vegetation responses to environmental conditions. The hourly visualization enables a better understanding of how each variable behaves at different times of the day, providing valuable insights for modeling atmospheric and ecological processes as well as for subsequent analyses of interdependence among these variables.

**Table 4.** Descriptive statistics of the eight time series, with $N = 75386$ observations.

| | $NEE$ (umolm$^{-2}$s$^{-1}$) | $H$ (Wm$^{-2}$) | $LE$ (Wm$^{-2}$) | $Rg$ (Wm$^{-2}$) | $rH$ (%) | $T_{air}$ °C | $T_{soil}$ °C | $VPD$ (hPa) |
|---|---|---|---|---|---|---|---|---|
| Mean | 0.54 | 36.0 | 78.5 | 224 | 73.8 | 26.4 | 29.8 | 12.6 |
| Median | 1.44 | 0.9 | 31.6 | 7.3 | 79.2 | 26.0 | 30.1 | 7.7 |
| Mode | -10.10 | 101.0 | 108.0 | 0.00 | 93.7 | 24.1 | 29.7 | 0.00 |
| $sd$ | 6.79 | 66.8 | 105.0 | 308 | 20.0 | 5.43 | 3.00 | 13.9 |
| Minimum | -35.00 | -188.0 | -244.0 | 0.00 | 11.8 | 1.3 | 18.2 | 0.00 |
| Maximum | 35.00 | 602.0 | 699.0 | 1270 | 99.8 | 42.6 | 39.6 | 83.5 |
| Skewness | -0.17 | 1.7 | 1.8 | 1.09 | -0.80 | -0.14 | -0.46 | 1.48 |
| Kurtosis | 3.50 | 2.5 | 2.8 | -0.235 | -0.293 | 0.190 | 0.296 | 1.99 |

### 3.1.2 Descriptive Statistics Considering Daily Seasonal Patterns

In this study, we consider the Net Ecosystem Exchange ($NEE$) as the central variable, as it directly reflects the dynamics of $CO_2$ exchange at the soil–atmosphere interface. Fig. 2 makes it possible to visualize the characteristic diurnal pattern of each variable and, crucially, to analyze the interactions and temporal lags among them, such as the lag of the temperature peak relative to maximum solar radiation and the ecosystem's photosynthetic response ($NEE$) to light availability. Taken together, these observed patterns and interactions form a biogeophysical signature that characterizes the intrinsic functioning and the dynamics of energy and mass exchange at the study site.

Fig. 2(a) shows that the highest $NEE$ values occur at night, between 7 p.m. and 11 p.m., reaching approximately $4$ μmolm$^{-2}$s$^{-1}$. This behavior is expected, since the absence of solar radiation inhibits photosynthesis, causing plants to stop carbon assimilation and release $CO_2$ through respiration. After this period, the mean $NEE$ values stabilize around 3.7 μmolm$^{-2}$s$^{-1}$ until sunrise, when solar radiation ($Rg$) begins to strike the surface (around 5 a.m.), reactivating photosynthesis. From that point on, there is a progressive decline in mean $NEE$ values, reaching a minimum around 1 p.m., with an average of approximately $-5.7$ μmolm$^{-2}$s$^{-1}$, coinciding with the $Rg$ peak shown in Fig. 2(b). This pattern clearly highlights $NEE$ as a sensitive





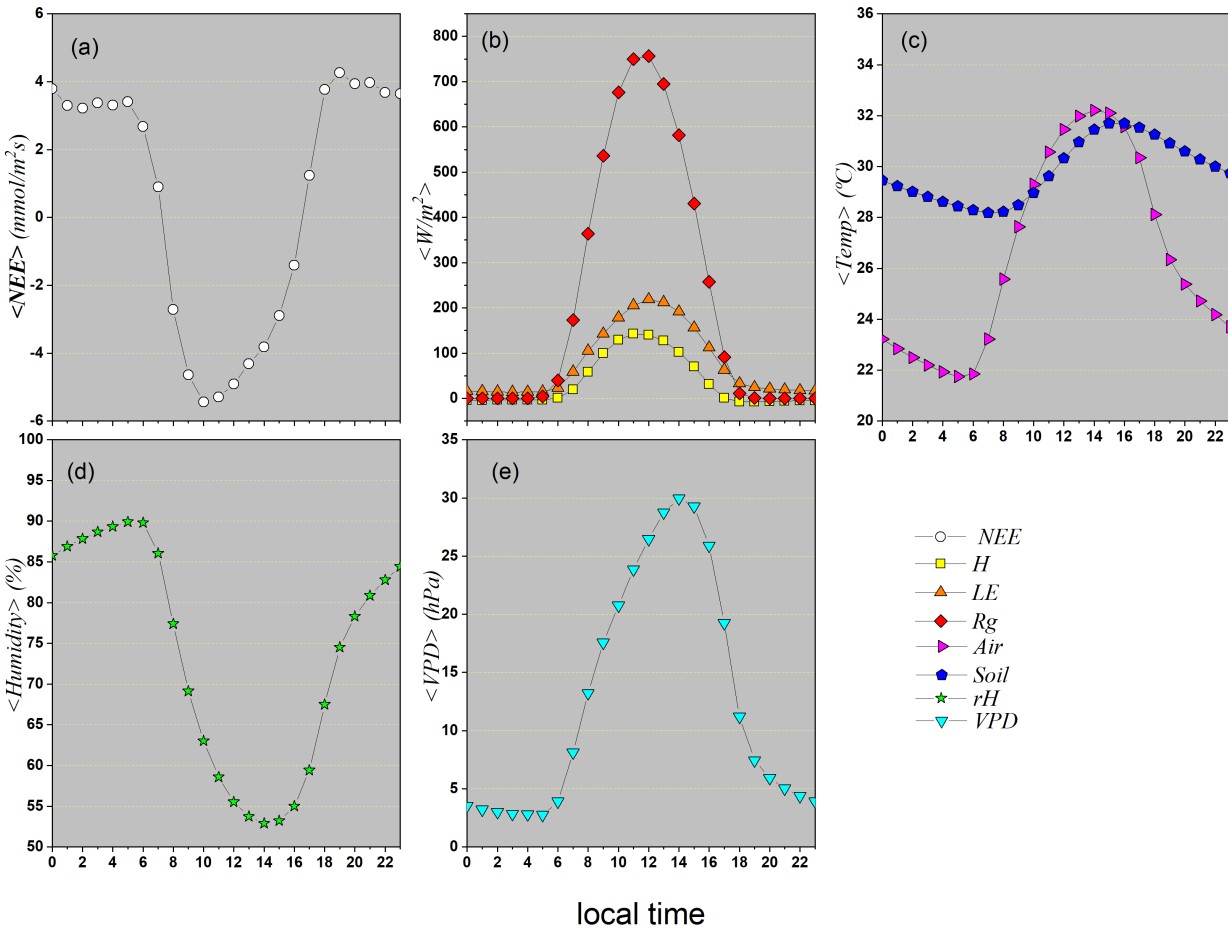

**Figure 2.** Mean values for each hour of the day (local time) of all eight variables.

indicator of the daily balance between CO$_2$ absorption and emission, strongly regulated by the alternation between daytime
photosynthesis and nighttime respiration.

Fig. 2(b) presents the average hourly profiles of global radiation ($Rg$), sensible heat flux ($H$), and latent heat flux ($LE$),
according to the definitions in Table 1. All three variables exhibit a near-normal distribution, with daytime peaks around 1
p.m., coinciding with peak solar incidence. The $Rg$ curve is symmetric and shows high kurtosis, reflecting a sharp solar
radiation peak concentrated between 12 p.m. and 1 p.m. The $LE$ profile, on the other hand, shows slight right-skewness,
indicating that values remain high for longer during the afternoon, even after $Rg$ starts to decline. This persistence in latent
heat flux can be explained by several factors, such as continued plant transpiration after peak radiation, soil water availability,
and the thermal inertia of biomass, which accumulates heat throughout the day and releases it gradually. Additionally, potential
nonlinearities in the relationship between $LE$ and vapor pressure deficit ($VPD$) may contribute to this behavior. Together,





these energy fluxes provide important insights into the energy exchange processes between the surface and the atmosphere, with direct implications for the mechanisms regulating the daily $CO_2$ cycle.

Fig. 2(c) presents the average hourly profiles of air temperature ($T_{air}$) and soil temperature ($T_{soil}$). Air temperature exhibits greater daily thermal amplitude compared to soil temperature, indicating a faster response of the atmosphere to variations in solar radiation throughout the day. This difference is attributed to the higher heat capacity and density of soil, which requires more energy to undergo noticeable temperature changes. As a result, the soil warms and cools more slowly, accumulating heat during the day and gradually releasing it at night. This thermal inertia plays an important role in modulating the surface

microclimate, contributing to ecosystem thermal stability and influencing other atmospheric and biogeochemical processes, such as evapotranspiration and surface energy balance.

Fig. 2(d) shows the average hourly profiles of relative humidity ($rH$) and vapor pressure deficit ($VPD$), revealing an inverse dynamic between these variables throughout the daily cycle. During the early morning hours, $rH$ reaches its highest values, close to 90%, due to lower temperatures and the reduced capacity of air to retain water vapor. As temperature rises throughout

the day, especially between 2 p.m. and 4 p.m., this capacity increases, resulting in a sharp drop in $rH$, reaching minimum values around 53%. Conversely, $VPD$ remains low overnight (about 3 hPa) and increases rapidly after sunrise, peaking around noon to early afternoon (about 29 hPa), when the atmosphere is hottest and driest. This mirrored relationship between $rH$ and $VPD$ is essential to understanding the control mechanisms of evapotranspiration, as $VPD$ is one of the main drivers of atmospheric evaporative demand on vegetation.

The daily-scale analysis reveals how the solar radiation cycle modulates the thermal and hydrological patterns of the ecosystem, directly influencing the behavior of net carbon flux ($NEE$). Increased global radiation ($Rg$) during the day raises air ($T_{air}$) and soil ($T_{soil}$) temperatures, reduces relative humidity ($rH$), and intensifies vapor pressure deficit ($VPD$), creating conditions that favor photosynthesis and, consequently, $CO_2$ absorption. At night, with the decline of $Rg$, air and soil masses cool down, humidity increases, and $VPD$ decreases, favoring plant respiration and $CO_2$ release. However, understanding the

mean daily behavior of these variables is not sufficient to capture broader variability patterns, such as those associated with meteorological events, seasonality, or persistent changes in climate conditions. Therefore, it is necessary to expand the analysis to a weekly scale, as presented in Fig. 3, in order to identify short-term trends and potential deviations from the typical diurnal pattern that may significantly influence energy, water vapor, and carbon fluxes in the ecosystem.

### 3.1.3 Descriptive Statistics Considering Weekly Seasonal Patterns

Fig. 3(a) presents the weekly mean of Net Ecosystem Exchange ($NEE$), revealing a clear seasonal pattern throughout the year. The negative values observed at the beginning of the year indicate a net absorption of $CO_2$ by vegetation, typical of the rainy season and the active growth phase. Between weeks 5 and 10, $NEE$ reaches its most negative values, reflecting peak photosynthetic activity driven by high water availability and intense solar radiation. From that point on, values become progressively less negative, eventually turning positive between weeks 20 and 30—coinciding with the peak of the dry season.

This behavior suggests a decline in photosynthesis, possibly due to water stress, and the predominance of plant respiration, resulting in net $CO_2$ emissions. Toward the end of the year, with the return of rainfall, $NEE$ again shows negative values,





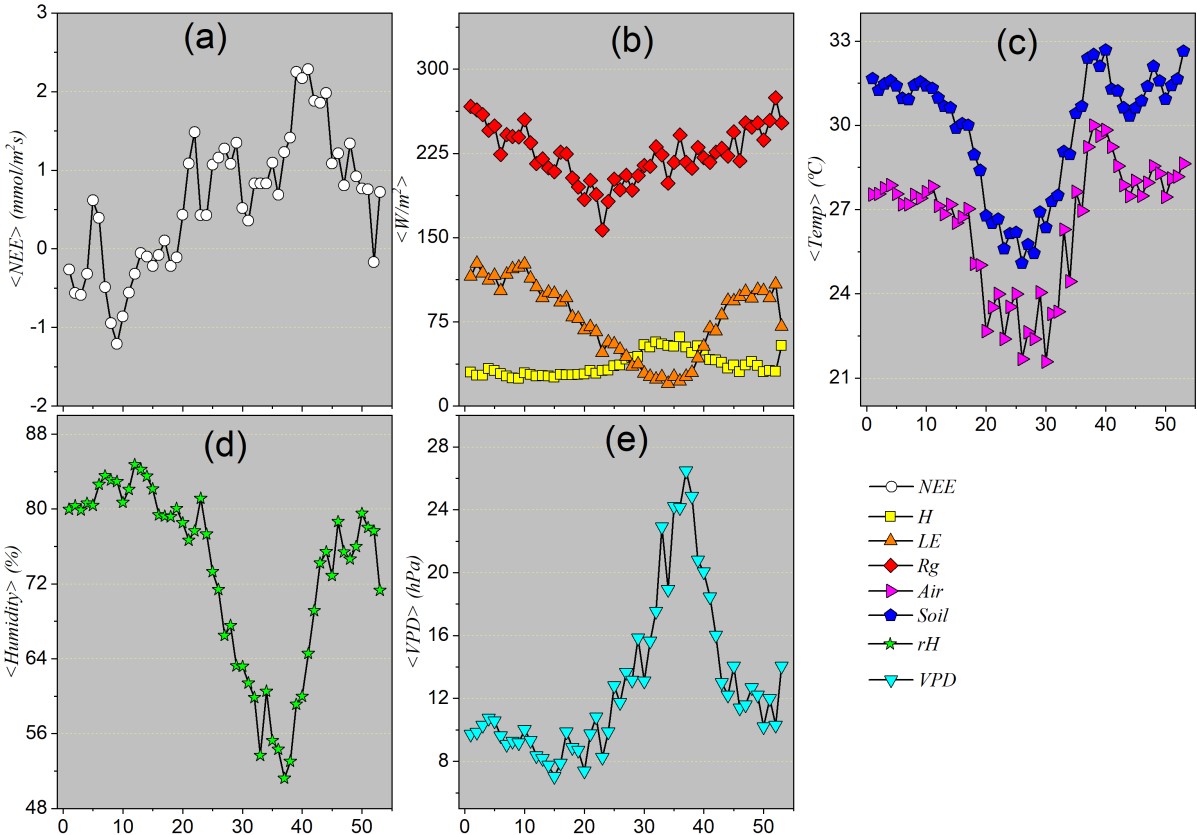

**Figure 3.** Weekly mean values for all eight variables under study.

indicating a resumption of carbon assimilation by vegetation. This pattern highlights the role of $NEE$ as a sensitive indicator of the interaction between climatic variables and ecosystem processes throughout the annual cycle.

The $LE$ curve shows its lowest values during the central part of the year (approximately between weeks 25 and 40). Since $LE$ is directly linked to water availability for evapotranspiration, these low values indicate a period of water stress or a dry season. During this phase, soil water is scarce, and plants close their stomata to avoid dehydration, drastically reducing transpiration. Consequently, little energy is used in this process. This interpretation is strongly supported by the sharp drop in Relative Humidity ($rH$) in panel (d) and the dramatic peak in Vapor Pressure Deficit ($VPD$) in panel (e) during the same period. A high $VPD$ indicates very dry air, which hampers plant transpiration.

In contrast, sensible heat flux ($H$) shows the opposite behavior of $LE$: it peaks between weeks 25 and 40, when $LE$ is at its minimum. This happens because the incident solar energy ($Rg$) that cannot be dissipated via evapotranspiration is converted into heat, increasing surface temperature and enhancing heat transfer to the atmosphere. The $H$ peak coincides with the highest





air temperatures, as shown in Fig. 3(c). This redistribution of available energy highlights the role of soil moisture in modulating the surface energy balance.

Global radiation ($Rg$), also shown in Fig. 3(b), follows a typical seasonal pattern of the annual solar cycle in the Southern Hemisphere: higher values at the beginning and end of the year (austral summer) and lower values during winter, reflecting the regional insolation regime.

    Weekly average air temperature ($T_{air}$) and soil temperature ($T_{soil}$), illustrated in Fig. 3(c), follow similar patterns, both with clear seasonality. Air temperature starts the year with higher values, reaches a minimum around weeks 25 to 30 (22 °C), and

rises again to about 32 °C by year-end. Soil temperature follows the same general pattern, with slightly higher mean values than air temperature and smaller variation amplitude due to its greater thermal inertia. This thermal behavior is relevant because it reveals a transition from anti-correlation to persistence between $T_{air}$ and $T_{soil}$ at the weekly scale, suggesting different dominating processes during distinct phases of the annual cycle.

    Fig. 3(d) shows that relative humidity ($rH$) is high at the beginning and end of the year (above 80%) and reaches minimum

values (53%) between weeks 25 and 30. Conversely, $VPD$, in Fig. 3(e), exhibits behavior opposite to $rH$: it is lower during humid periods (8 hPa) and peaks at the height of the dry season (25 hPa).

    In summary, the weekly averages analysis reveals the existence of two well-defined climatic regimes in the study region: a rainy season, concentrated in the early and late months of the year, characterized by higher $CO_2$ absorption (negative $NEE$), high relative humidity, low $VPD$ values, and intense evapotranspiration activity (high $LE$); and a dry season, in the middle of

the year, marked by net $CO_2$ release, decreased relative humidity, high $VPD$, reduced $LE$, and increased $H$. These patterns reflect the strong climatic control over energy and carbon fluxes in the Pantanal ecosystem.

### 3.1.4 Descriptive Statistics Considering Monthly Seasonal Patterns

To consolidate the understanding of the observed seasonal cycle, following the analysis of hourly and weekly scales, we now turn to the evaluation of the average monthly behavior of the variables. The monthly mean is an important tool as it smooths

out short-term fluctuations highlighted in the weekly averages, allowing larger-scale climatic patterns — such as the onset, peak, and end of the wet and dry seasons — to become more evident. This data aggregation helps reduce statistical noise and enhance the dominant seasonal signal in a more robust manner. Thus, monthly analysis provides a macroscopic and integrative perspective, essential for accurately characterizing the climatic periods that govern ecosystem dynamics throughout the year.

    Fig. 4 shows the monthly means of the atmospheric and ecosystem variables analyzed. Net Ecosystem Exchange ($NEE$)

displays negative values in the first months of the year, indicating net $CO_2$ uptake by vegetation. From March onward, there is a gradual increase in these values, which become positive in June and peak in September. Then, the curve decreases again, returning to negative values in December. This pattern reflects the ecosystem's annual productivity cycle: net $CO_2$ absorption mainly occurs during the rainy season, when vegetation is most active, while net emissions (positive $NEE$ values) between May and September indicate a period of reduced photosynthetic activity, characteristic of the dry season.

Global radiation ($Rg$) peaks during the summer (December to February) and reaches its minimum in winter (May to July), with monthly mean values around 175 W m$^{-2}$ between June and July. Sensible heat flux ($H$) generally follows the same





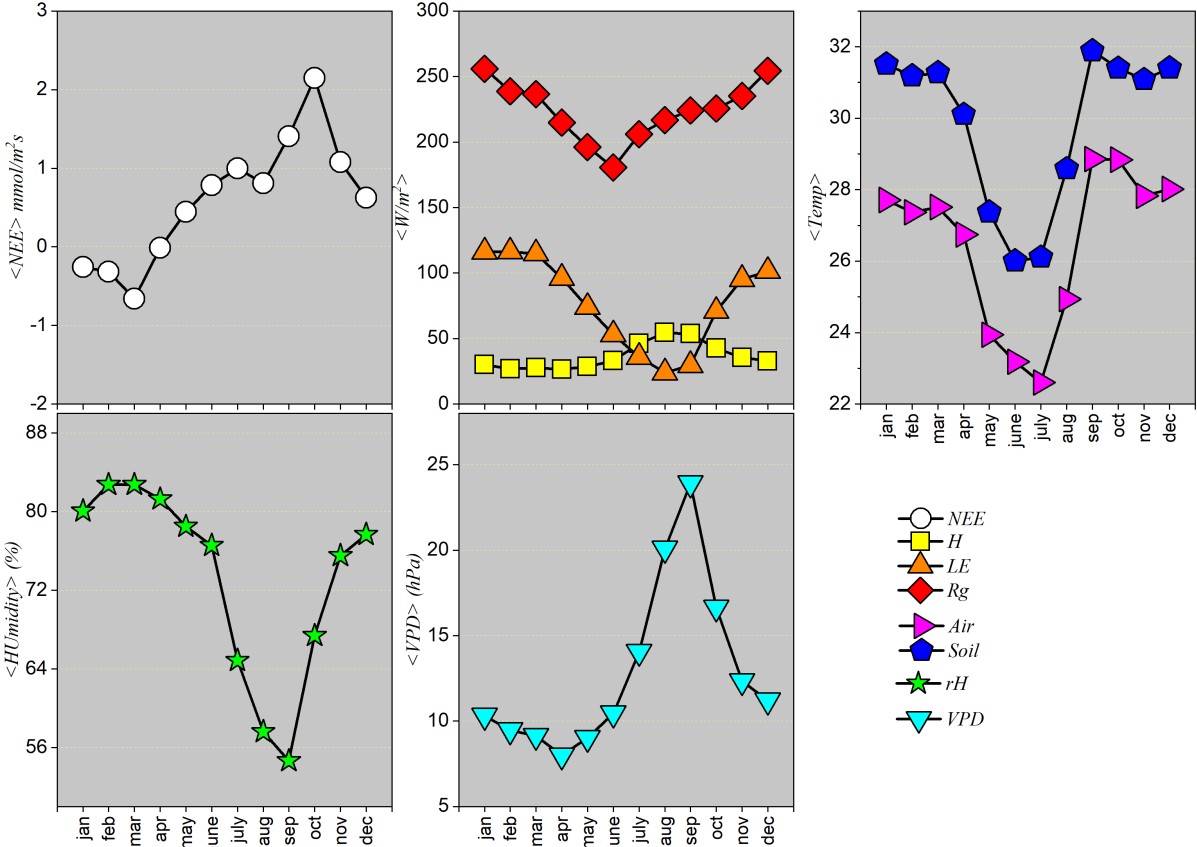

**Figure 4.** Monthly mean values for all variables under study.

pattern as $Rg$, with high values during the warm/wet season and a significant drop during the dry season. Latent heat flux ($LE$) also follows a similar trend, but with nuances related to water availability: in some months of the dry season (July, August, and September), $LE$ is lower than $H$, reflecting the limitation of water for evapotranspiration. During the rainy season, $LE$ increases, showing that energy partitioning between $H$ and $LE$ is sensitive to the water balance: higher $LE$ during wet periods and higher $H$ during dry periods, when energy is predominantly used for warming the environment.

Air temperature ($T_{air}$) starts the year with average values above $26\,°C$ (January to March), decreasing to its annual minimum in June–July (below $24\,°C$), and rising again in the second half of the year, with a new peak in September. Soil temperature ($T_{soil}$) follows a similar pattern to $T_{air}$, but with slightly higher values and lower thermal amplitude, reflecting the thermal inertia of the soil. These temperature patterns confirm the alternation between warm and cool seasons in the Pantanal, with the cooler period coinciding with the dry season.





Relative humidity ($rH$) shows high values from January to April (above 80%), gradually decreasing during the dry season, reaching minimum values between August and September (55–60%). Humidity begins to rise again with the onset of the rainy season in October. Conversely, vapor pressure deficit ($VPD$) exhibits behavior opposite to $rH$: it begins the year with values between 8 and 10 hPa, increases significantly during the dry season, peaking at around 24 hPa in September, and then decreases with the return of the rains. The high $VPD$ during the dry season indicates extremely dry air, increasing the evaporative demand on vegetation and promoting water stress.

Fig. 4 clearly summarizes the annual climatic and hydrological cycle of the Pantanal, highlighting two well-defined seasons: - **Rainy season (October to April)**: marked by negative $NEE$ (net $CO_2$ uptake), high temperatures, high global radiation, high $LE$, high relative humidity, and low $VPD$. - **Dry season (June to September)**: characterized by positive $NEE$ (net $CO_2$ emission), lower temperatures, reduced global radiation, dominance of $H$ over $LE$, low relative humidity, and high $VPD$.

With the monthly analysis confirming a well-defined seasonal pattern, the next step is to group the data into a quarterly scale. This approach synthesizes the ecosystem's average states during the peak of each season, reducing intra-seasonal variability and facilitating comparisons between climatic macroperiods — such as the peak of the rainy season and the core of the dry season. Therefore, quarterly aggregation will be essential for deepening the functional characterization of the Pantanal ecosystem's annual cycle.

Having the analysis of monthly averages consolidated the existence of a well-defined seasonal pattern, the next step consists of grouping these data on a quarterly scale. This approach allows one to go beyond month-to-month variability and focus on characterizing the year's major climatic periods. Quarterly aggregation functions as a synthesis tool that quantifies the average state of the ecosystem during the peak of each season, enabling a direct and robust comparison between the fundamentally distinct periods that comprise the annual cycle, such as the height of the rainy season and the core of the dry season.

**Descriptive Statistics Considering Seasonal Quarterly Patterns**

Quarterly averages represent an integrated synthesis of microclimatic conditions and the fluxes of energy and matter throughout the main seasonal phases of the year. Unlike the sequential analysis provided by monthly means, the quarterly approach offers representative "snapshots" of the ecosystem's average behavior during the peak of the rainy season, the dry season, and transitional periods. This aggregation scale allows not only for the identification but also the quantification of the intensity and magnitude of the dominant processes in each season. For example, it becomes possible to highlight the predominance of latent heat flux ($LE$) during the wet season, in contrast with the dominance of sensible heat flux ($H$) during the dry season—allowing the amplitude of local seasonality to be defined based on concrete average values.

From a climatic perspective, the quarters analyzed do not constitute arbitrary calendar divisions but rather representative periods of the distinct phases of the hydrological cycle in the studied region. The first quarter (January–February–March) generally corresponds to the peak of the rainy season, characterized by high water availability, elevated relative humidity, and intense evapotranspirative activity. The second quarter (April–May–June) represents the transition from the rainy to the dry season, with a gradual decline in precipitation and shifts in energy and moisture patterns. The third quarter (July–August–September) represents the core of the dry season, with minimal precipitation, low relative humidity, high vapor pressure deficit ($VPD$) val-



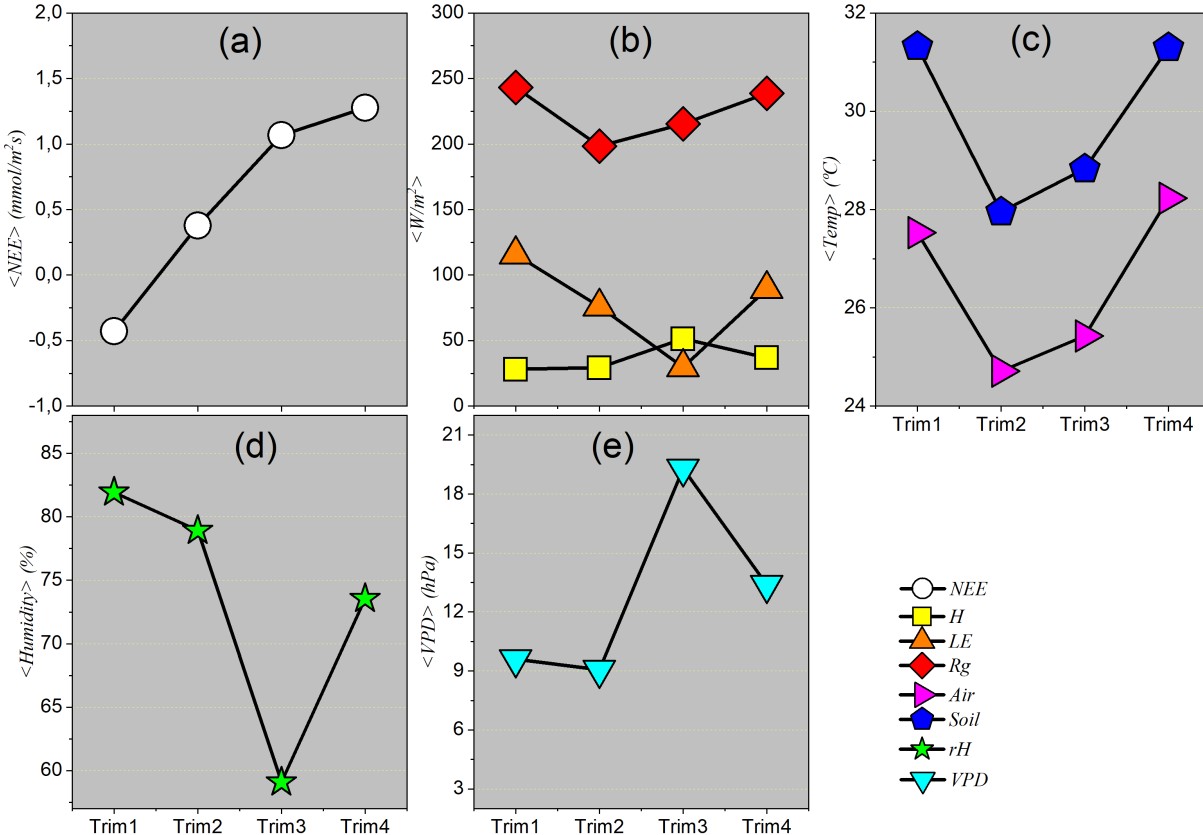

**Figure 5.** Quarterly average values for all variables under study.

ues, and significant water stress on vegetation. Finally, the fourth quarter (October–November–December) marks the gradual return of the rainy season, with increasing precipitation and a relief of dry conditions.

Fig. 5 presents the quarterly means of the analyzed variables. It can be observed that the average behavior remains consistent with the patterns identified in the monthly averages, reinforcing the reliability of seasonal cycles. However, the quarterly anal-
340  ysis more clearly highlights the periods of maximum and minimum activity in climatic and ecosystem processes, functioning as an effective tool for characterizing the seasonal regimes that shape the environmental dynamics of the Pantanal.

## 3.2  Results from the DFA Method

We now proceed with the autocorrelation analysis of the time series using the Detrended Fluctuation Analysis (DFA) method, as described in Section 2.2.1. This method is a robust tool for investigating long-range correlations in non-stationary time




series, allowing us to quantify the intrinsic "memory" of microclimatic processes — something that traditional autocorrelation methods do not adequately capture.

Fig. 6 presents the fluctuations calculated by the DFA method for each of the eight variables analyzed. The interpretation of these results is based on the values of the $\alpha$ exponents, listed in Table 5. This table provides the estimated autocorrelation coefficients at different temporal scales, based on the original series measured every 30 minutes (average of records captured at 10 Hz). Considering the seasonality discussed in the previous sections, we adopted four ranges of temporal scale for estimating $\alpha$:

1. Daily scale: $4 \leq n \leq 48$ (corresponding to the first section of Fig. 6, delimited by vertical dashed lines);

2. Weekly scale: $48 < n \leq 336$ (for some variables, such as temperatures, two distinct behaviors are observed within this range);

3. Monthly scale: $336 < n \leq 1440$;

4. Quarterly scale: $n > 1440$.

The values of the $\alpha$ exponents for each variable and scale are presented in Table 5. We begin the analysis with the behavior of $NEE$. At the daily scale ($4 \leq n \leq 48$), $\alpha = 0.96$ was obtained, indicating strong short-term correlation — that is, the $CO_2$ flux on one day is strongly influenced by the conditions of the previous day, such as solar radiation, temperature, and vegetation physiological activity. At the weekly scale ($48 < n \leq 336$), $\alpha = 0.24$ reveals significant anti-correlation, possibly related to rhythmic fluctuations (such as weekly human activity cycles) or ecosystem saturation mechanisms (e.g., after a peak of $CO_2$ uptake, compensation occurs in the following days). At the monthly scale ($336 < n \leq 1440$), $\alpha = 0.85$ indicates moderate persistence, consistent with the influence of gradual seasonal trends. At the quarterly scale ($n > 1440$), $\alpha = 1.07$ indicates long-range persistence, reflecting broad seasonal patterns and possible ecosystem responses to larger-scale climatic variations.

For the other variables, the DFA analysis revealed the following general patterns:

– **Daily scale** ($4 \leq n \leq 48$): $\alpha = 0.96$, indicating strong persistence. This suggests that daily variations in $NEE$ are highly influenced by previous-day conditions such as solar radiation, temperature, and vegetation physiology.

– **Weekly scale** ($48 < n \leq 336$): $\alpha = 0.24$, indicating strong anti-correlation. This may reflect ecological saturation patterns or recurrent external interferences such as weekly human activity.

– **Monthly scale** ($336 < n \leq 1440$): $\alpha = 0.85$, suggesting persistent correlation. This trend may be associated with gradual seasonal changes, such as increased photosynthesis in spring.

– **Quarterly scale** ($n > 1440$): $\alpha = 1.07$, indicating strong long-term persistence, possibly related to broad seasonal variations and prolonged climate trends.

The analysis of the other variables reveals a pattern consistent with ecosystem physiology:





**Table 5.** DFA Exponent

- **Daily scale**: Most variables ($H$, $Rg$, $LE$, $rH$, $T_{air}$, $T_{soil}$, $VPD$) show $\alpha > 1.00$, indicating strong daily persistence and trend, as expected due to natural diurnal cycles (solar radiation, temperature, humidity).

- **Weekly scale**: A predominant trend toward anti-correlation is observed ($\alpha < 0.50$), suggesting that weekly variations are often followed by compensations in the opposite direction.

- **Monthly scale**: The series exhibit moderate correlation ($0.50 < \alpha < 1.00$), reflecting seasonal persistence.

- **Quarterly scale**: Most variables return to a strong trend ($\alpha > 1.00$), associated with broad seasonal dynamics.

Among the variables analyzed, global radiation ($Rg$) and $NEE$ stand out for not presenting $\alpha$ exponents greater than 1.00 across all scales, unlike the others. $NEE$ shows particularly complex behavior, with strong daily persistence, weekly anti-correlation, and long-term persistence. Meanwhile, $Rg$ exhibits behavior close to white noise on the monthly scale, reflecting high meteorological variability and the more stochastic nature of incoming radiation.

The application of DFA allowed us to clearly characterize the degree of persistence and the presence of long-range correlations in each time series, revealing dynamic patterns not readily detectable by conventional methods. However, since it is a univariate approach, DFA is not capable of capturing interdependencies between distinct variables. To advance in understanding the relationships between ecosystem carbon flux ($NEE$) and the microclimatic factors that influence it, we apply an extension of DFA: the Detrended Cross-Correlation Analysis (DCCA), presented in the next section. DCCA allows us to estimate the strength and direction of correlations between pairs of non-stationary time series, using the $\rho_{\text{DCCA}}$ coefficient.

1. Daily time scale, where $4 \leq n \leq 48$ (these windows correspond to the first section of Fig. 6, see vertical dashed lines in the figure);

2. Weekly time scale, where $48 < n \leq 336$ (note that for temperatures, there are two values for the weekly scale, reflecting a change in behavior within this scale);

3. Monthly time scale, where $336 < n \leq 1440$;

4. Quarterly time scale, $n > 1440$.

### 3.3 Results for the $\rho_{\text{DCCA}}$ Coefficient

In this section, we specifically investigate the correlations between the net ecosystem carbon flux ($NEE$) and the other micrometeorological variables. Fig. 7 shows the $\rho_{\text{DCCA}}$ values obtained for each pair involving $NEE$. This analysis allows us to explore how different environmental factors influence the patterns of $CO_2$ exchange across multiple temporal scales, revealing the dynamics that govern the relationship between the ecosystem and the atmosphere.





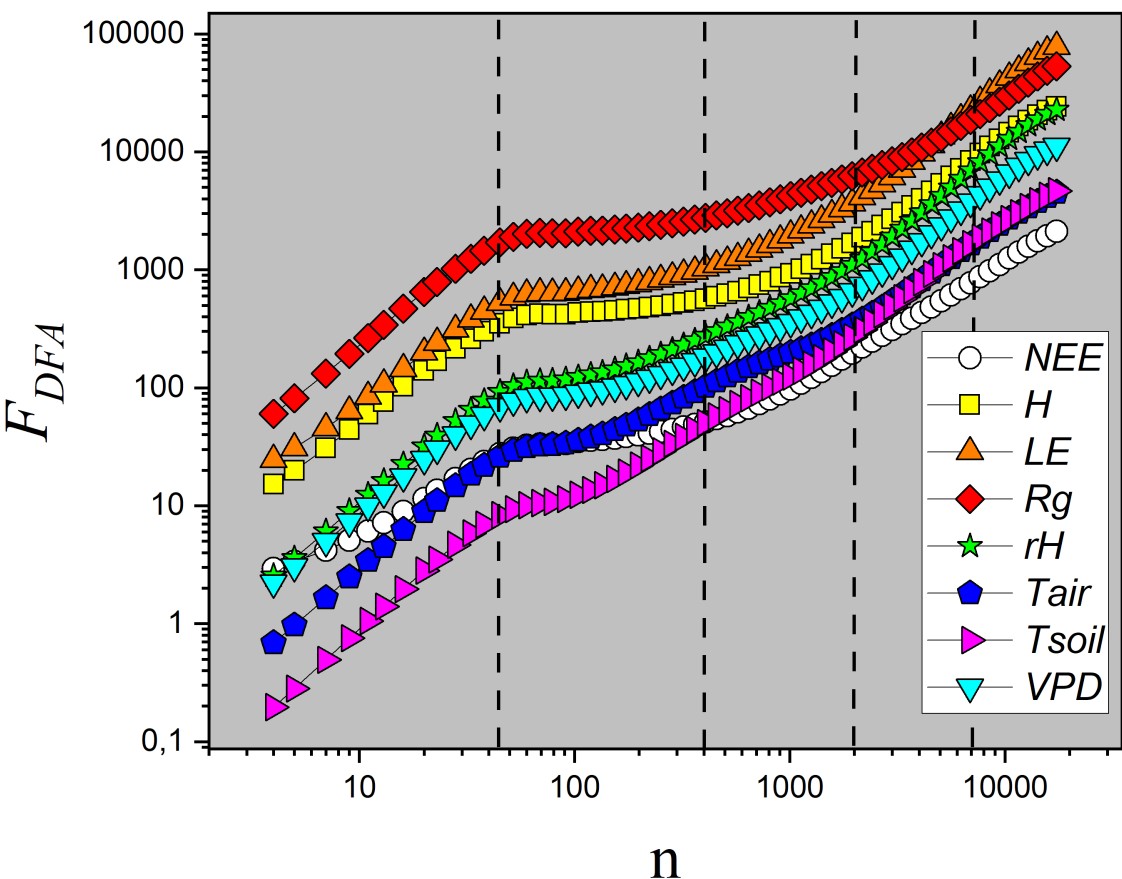

**Figure 6.** Plot of $F_{DFA}$ as a function of the temporal scale $n$ for the eight variables. In this analysis, $n$ represents the number of observation windows, each with a 30-minute duration. The vertical lines indicate the time scales of interest: daily ($n = 48$), weekly ($n = 336$), monthly ($n = 1440$), and quarterly ($n = 4320$).

Fig. 7(a) shows the cross-correlations between $NEE$ and the sensible heat flux ($H$, yellow squares), and between $NEE$ and the latent heat flux ($LE$, orange triangles). For $NEE \times H$, a weak anti-correlation is observed at small temporal scales, with $\rho_{DCCA} \approx -0.15$. The curve decreases to a minimum of approximately $\rho_{DCCA} \approx -0.6$ at the daily scale, indicating a moderate anti-correlation. From this scale onward, the correlation gradually becomes less negative, approaching zero at the monthly scale and reaching $\rho_{DCCA} \approx 0.25$ at the quarterly scale, characterizing a weak correlation. In the case of $NEE \times LE$, there is also a weak anti-correlation at small scales, with $\rho_{DCCA} \approx -0.15$, which intensifies to a minimum of approximately $-0.8$ at the daily scale, indicating strong anti-correlation. At the weekly scale, the correlation is moderate ($\rho_{DCCA} \approx -0.4$),





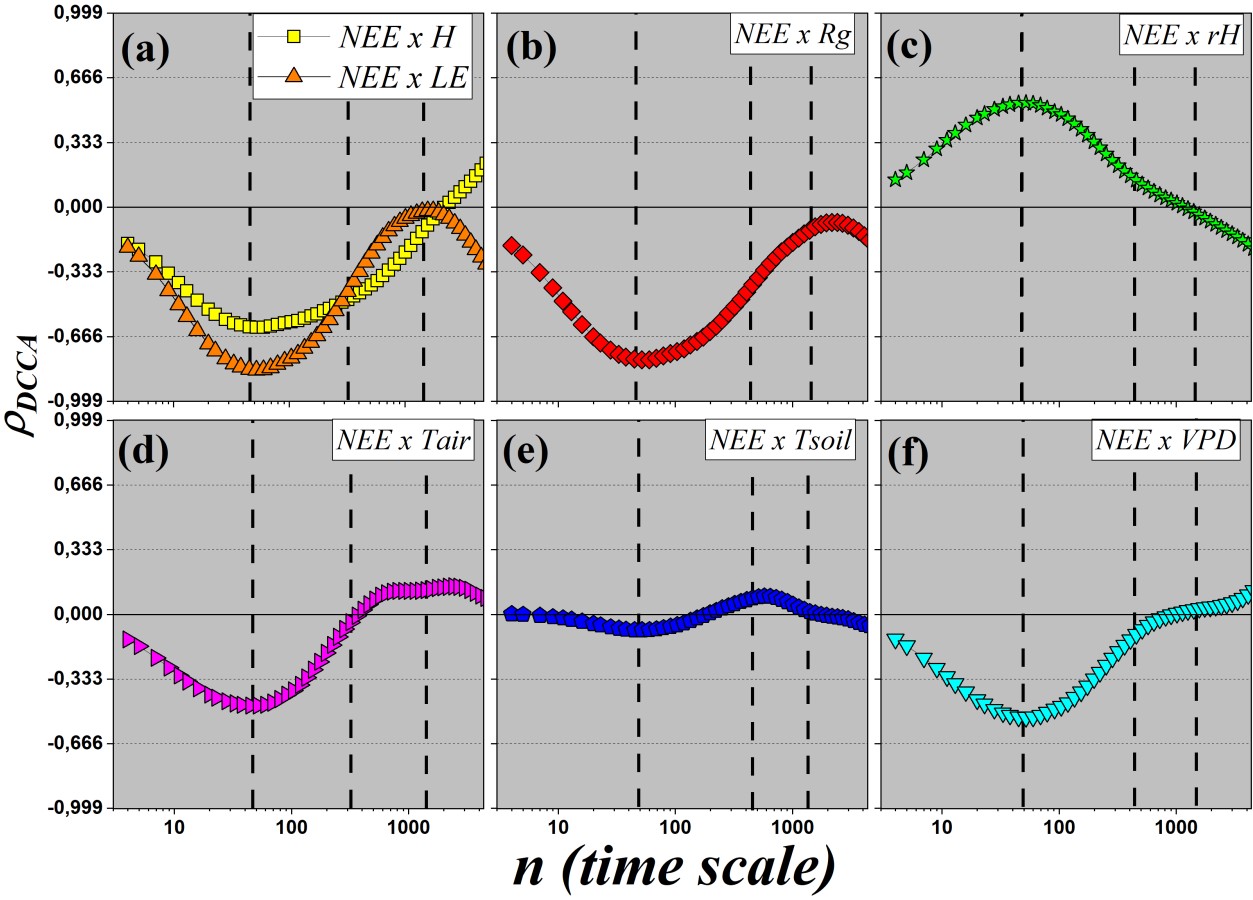

**Figure 7.** DCCA cross-correlation coefficient ($\rho_{DCCA}$) as a function of the time scale $n$. Each panel displays the cross-correlation between the carbon flux ($NEE$) and one of the following micro-meteorological variables: (a) sensible ($H$) and latent ($LE$) heat fluxes; (b) global radiation ($Rg$); (c) relative air humidity ($rH$); (d) air temperature ($Tair$); (e) soil temperature ($Tsoil$); and (f) vapor pressure deficit ($VPD$).

and tends toward zero at the monthly scale. The curve exhibits an inflection at the weekly scale, with upward concavity up
410   to the daily scale and downward concavity up to the quarterly scale, where $\rho_{DCCA} \approx -0.3$. The presence of strong daily anti-correlation between $NEE$, $H$, and $LE$ suggests a complex behavior. Although photosynthesis and evapotranspiration are coupled in the diurnal cycle, at larger scales the correlation weakens, reflecting energy partitioning and the action of other limiting factors.

   Fig. 7(b) shows the cross-correlation between $NEE$ and global radiation ($Rg$). A pattern similar to $LE$ is observed, with
415   $\rho_{DCCA} \approx -0.2$ at small scales, decreasing to approximately $-0.7$ at the daily scale (strong anti-correlation). The correlation becomes less negative at larger scales, passing through $-0.4$ at the weekly scale (moderate anti-correlation) and approaching zero at the monthly scale. At the quarterly scale, values return to about $-0.2$, indicating weak anti-correlation. The strong daily





anti-correlation with $Rg$ reflects the complex balance between radiation and ecosystem carbon dynamics: although radiation drives photosynthesis, high $Rg$ often coincides with elevated $VPD$ and respiration, which can lead to opposite persistence patterns in $NEE$, particularly under conditions of water stress or other environmental constraints.

Fig. 7(c) shows the correlation between $NEE$ and relative humidity ($rH$), which is the only variable that shows a positive correlation with $NEE$. At small scales, the correlation is weak ($\rho_{DCCA} \approx 0.2$), increasing to a maximum of $\rho_{DCCA} \approx 0.55$ at the daily scale, indicating moderate correlation. At the weekly scale, the correlation returns to weak values and becomes null at the monthly scale. At the quarterly scale, there is a slight anti-correlation ($\rho_{DCCA} \approx -0.25$). The moderate positive daily correlation between $NEE$ and $rH$ is ecophysiologically consistent, since higher relative humidity reduces water stress, favoring photosynthesis and increasing $CO_2$ uptake.

Fig. 7(d) presents the results of $\rho_{DCCA}$ between $NEE$ and air temperature ($T_{air}$). At small scales, a weak anti-correlation is observed ($\rho_{DCCA} \approx -0.2$), with a minimum of $-0.45$ at the daily scale (moderate anti-correlation). At the weekly scale, the correlation becomes nearly null, and at larger scales the values slightly increase, remaining weak and close to zero. The transition from moderate anti-correlation to weak or null correlation at larger scales may reflect the action of an optimal temperature range for photosynthesis, where extreme temperatures hinder $CO_2$ sequestration.

In Fig. 7(e), the correlation between $NEE$ and soil temperature ($T_{soil}$) is presented. A correlation close to zero is observed for most scales, with discrete variations: a slight dip at the daily scale and a small increase at the weekly scale. In both cases, the correlation remains weak, with no apparent significance. These results indicate that soil temperature exerts little direct influence on the long-range fluctuations of $NEE$, playing a secondary role compared to other variables.

Finally, Fig. 7(f) shows the cross-correlation between $NEE$ and vapor pressure deficit ($VPD$). At small scales, a weak anti-correlation is observed ($\rho_{DCCA} \approx -0.15$), which intensifies to $\rho_{DCCA} \approx -0.55$ at the daily scale (moderate anti-correlation). At the weekly scale, the correlation weakens ($\rho_{DCCA} \approx -0.15$), becoming null at the monthly scale. At the quarterly scale, a weak positive correlation appears ($\rho_{DCCA} \approx 0.25$). The moderate anti-correlation observed at the daily scale aligns with ecophysiological expectations, as high $VPD$ values indicate greater evaporative demand, increasing water stress and reducing photosynthetic activity, consequently lowering $CO_2$ uptake.

## 4 Conclusion

This study demonstrates that the relationship between carbon flux and climate in the Pantanal is multifaceted and strongly scale-dependent. By applying DFA and DCCA methodologies, we quantified the temporal memory and strength of correlations, revealing a hierarchy of control mechanisms acting at daily, weekly, and seasonal scales. The Pantanal's function as a carbon sink or source is therefore not static, but emerges from these scale-dependent interactions.

A key finding of our analysis is the detection of weekly anti-persistence in the Net Ecosystem Exchange ($NEE$) signal, where high values are likely to be followed by low values and vice versa. This sub-monthly regulation is rarely considered in ecosystem models, yet it may arise from soil moisture depletion, delayed physiological responses, or respiratory rebounds after





wetting. Incorporating these short-term processes could refine predictive capacity by acknowledging that recovery rhythms are an intrinsic part of ecosystem regulation, rather than noise around daily or seasonal drivers.

     At the daily scale, fluxes respond predictably to solar radiation and evaporative demand, while seasonal dynamics follow the hydrological pulse that shifts the system from a sink in the wet season to a source in the dry season. Similar multi-scale behavior has been reported elsewhere, although the underlying mechanisms vary. In tropical seasonal forests, weekly respiration pulses

often follow rewetting events (Zhang et al., 2010); in tropical peat swamp forests, disturbance and drainage rapidly convert systems from sinks to sources (Hirano et al., 2012); and in temperate ecosystems, long-term eddy covariance records document memory effects across scales (Desai et al., 2022). Even at continental scales, as during the 2015–2016 El Niño, drought-induced lags in carbon flux recovery were driven by the combined effects of atmospheric aridity and water storage deficits (Liu et al., 2024). Together, these examples place the Pantanal within a broader picture where ecosystem resilience arises from short-lived

rebounds embedded in longer-term climatic constraints.

     Comparisons with other biomes further support this interpretation. In a tropical seasonal forest, woody tissue respiration was estimated at about 10% of gross primary productivity (GPP), with strong dependence on leaf area index (LAI) (Meir and Grace, 2002). Low carbon use efficiency (CUE) observed in Amazonian forests, only about 30%, indicates that much of the assimilated carbon is rapidly released through respiration (Chambers et al., 2004). These structural and metabolic constraints

help explain the capacity for rapid ecosystem-scale fluctuations we observe in the Pantanal. The weekly anti-persistence we report is also consistent with current views of drought resilience, which emphasize not only resistance but also recovery and regulation across multiple temporal scales (Lu et al., 2025).

     We also recognize that carbon balance measurements in tropical ecosystems remain challenging. In an Amazonian forest in Pará, for example, annual estimates of carbon balance were shown to be highly sensitive to the treatment of nighttime eddy

covariance data, highlighting the importance of independent biometric checks (Miller et al., 2004). In tropical peat swamps, water table fluctuations alter the balance between peat decomposition and methane emissions, underscoring the need for long-term monitoring to capture both carbon loss and greenhouse gas trade-offs (Darusman et al., 2022). Against this backdrop of ecological complexity and methodological uncertainty, the detection of weekly anti-persistence in the Pantanal provides a concrete and quantifiable marker of ecosystem resilience.

In summary, identifying weekly anti-persistence in $NEE$ reveals new aspects of how tropical floodplains regulate carbon exchange. This finding has practical implications: models of regional climate should incorporate sub-monthly regulation tied to hydrological pulses and short-lag physiology, while conservation strategies must prioritize the preservation of natural flooding regimes and the mitigation of meteorological extremes. Safeguarding these processes is essential for maintaining the Pantanal's intricate and globally relevant role in the carbon cycle.



**Appendix A: Figures**

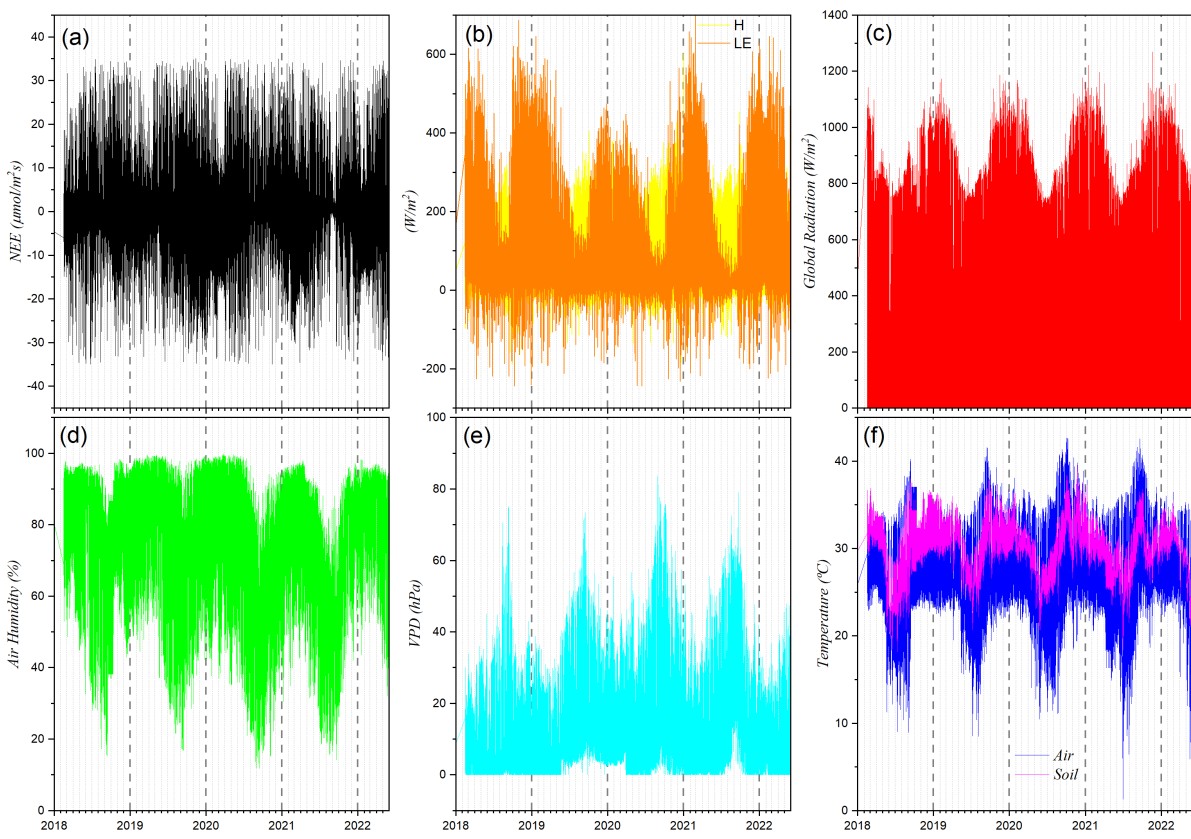

**Figure A1.** Time series of (a) Net Ecosystem Exchange ($NEE$), (b) sensible ($H$) and latent ($LE$) heat fluxes, (c) global radiation, (d) relative air humidity, (e) vapor pressure deficit ($VPD$), and (f) air and soil temperature for the Baía das Pedras site (Northern Pantanal) as a function of time.




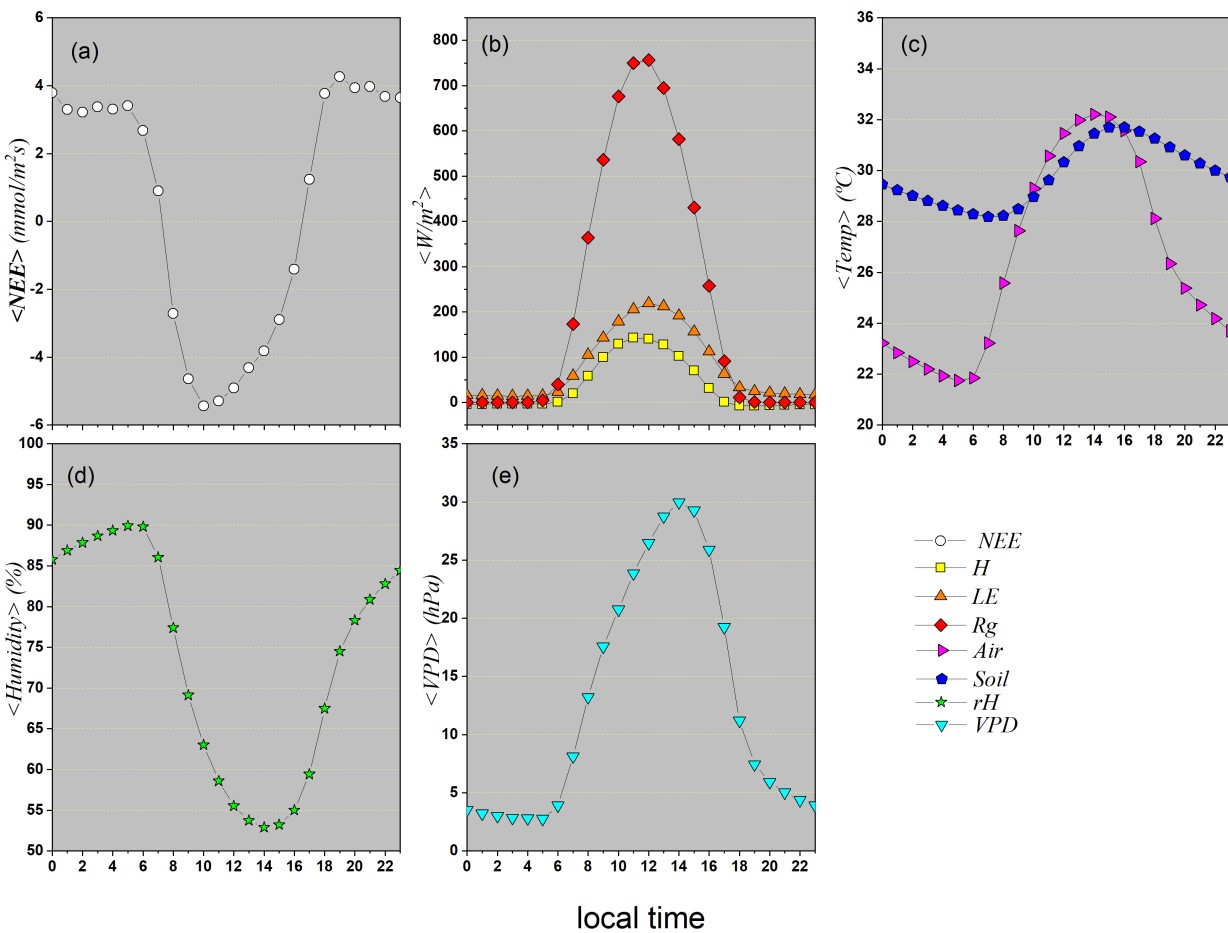

**Figure A2.** Mean values for each hour of the day (local time) of all eight variables.



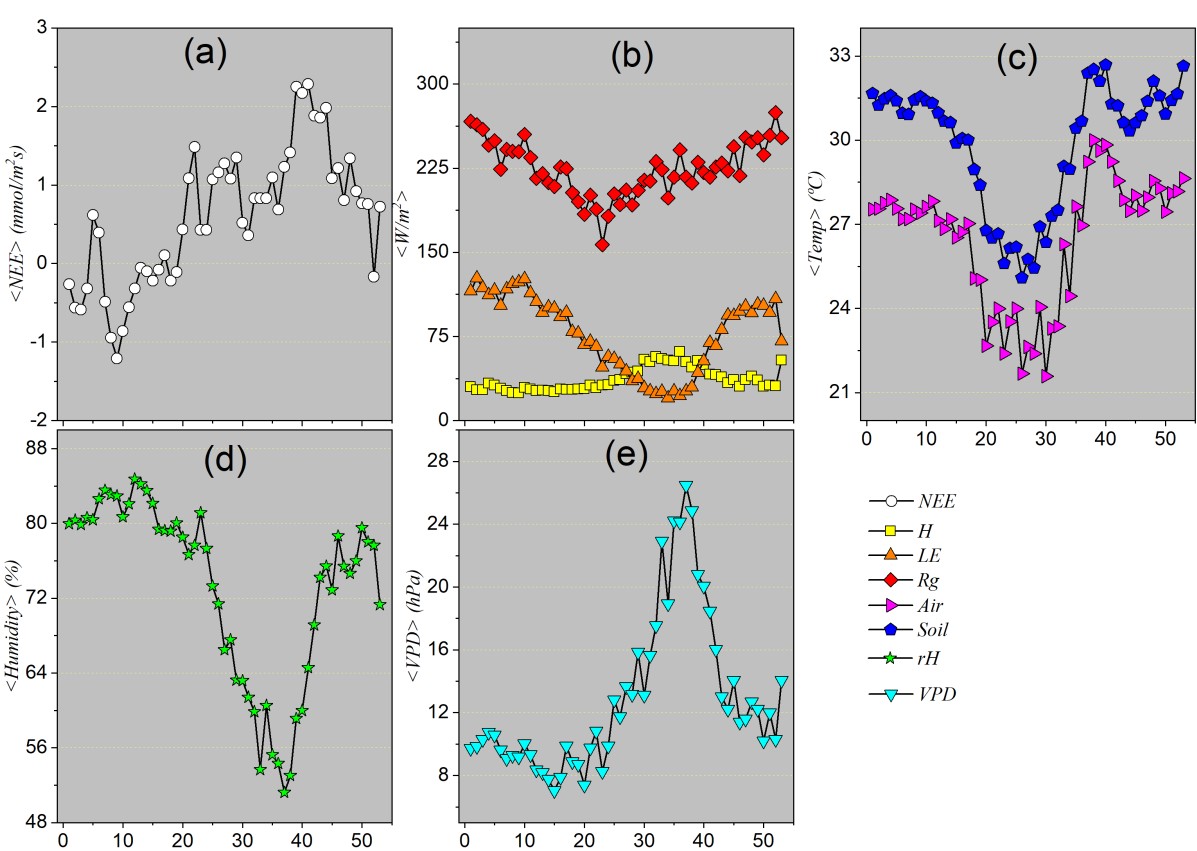

**Figure A3.** Weekly mean values for all eight variables under study.





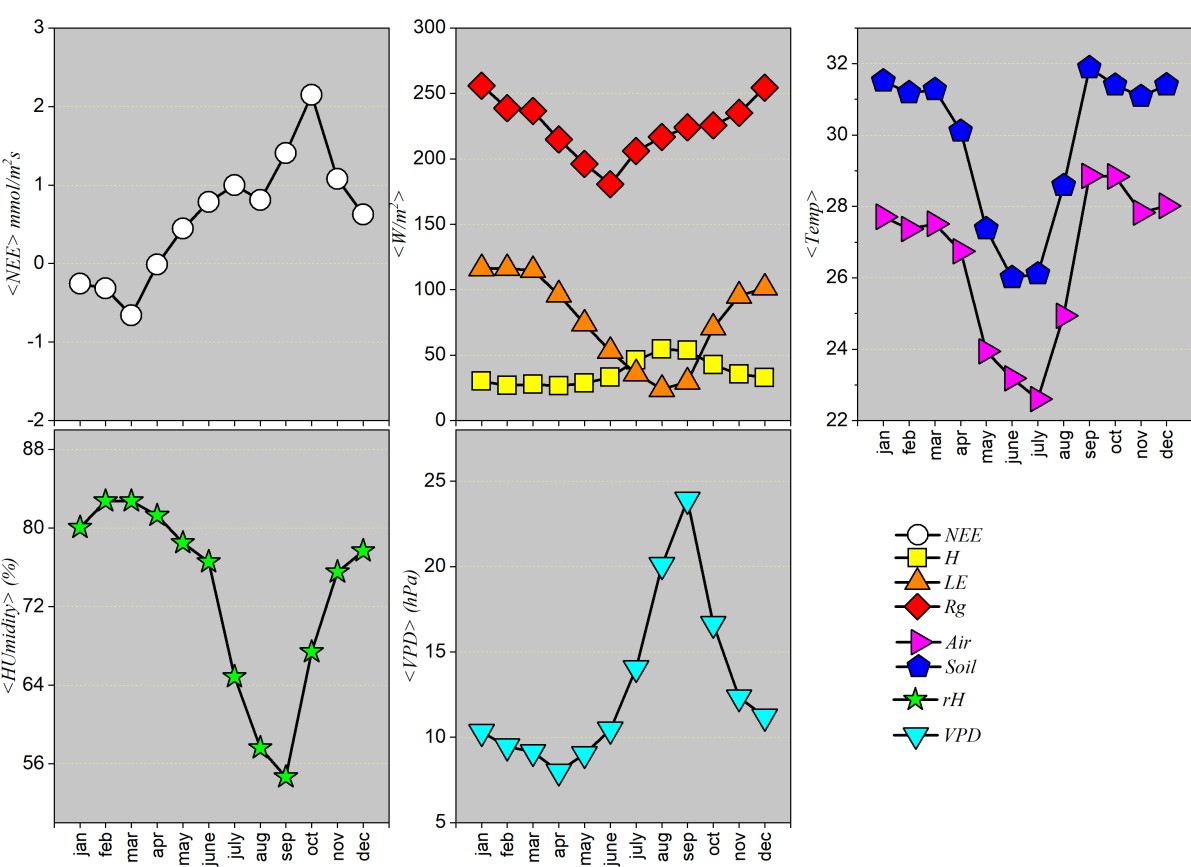

**Figure A4.** Monthly mean values for all variables under study.



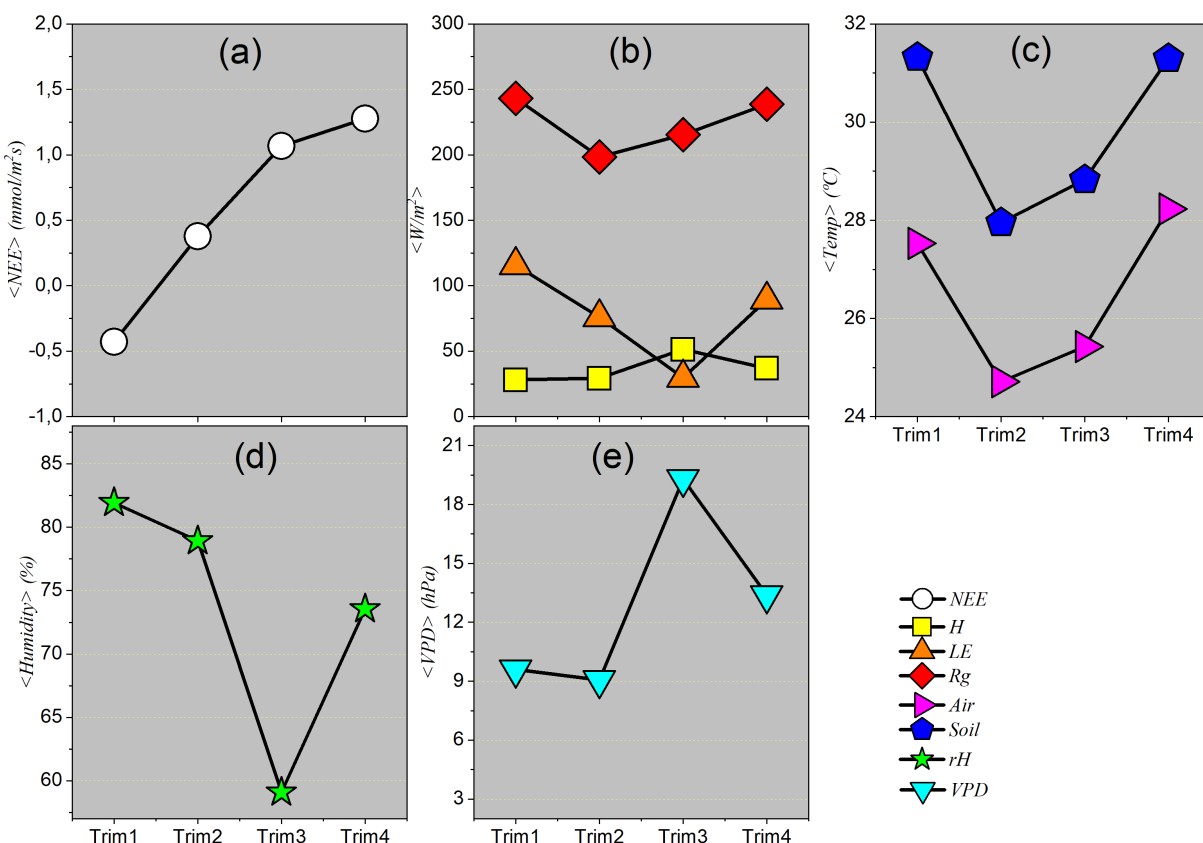

**Figure A5.** Quarterly average values for all variables under study.




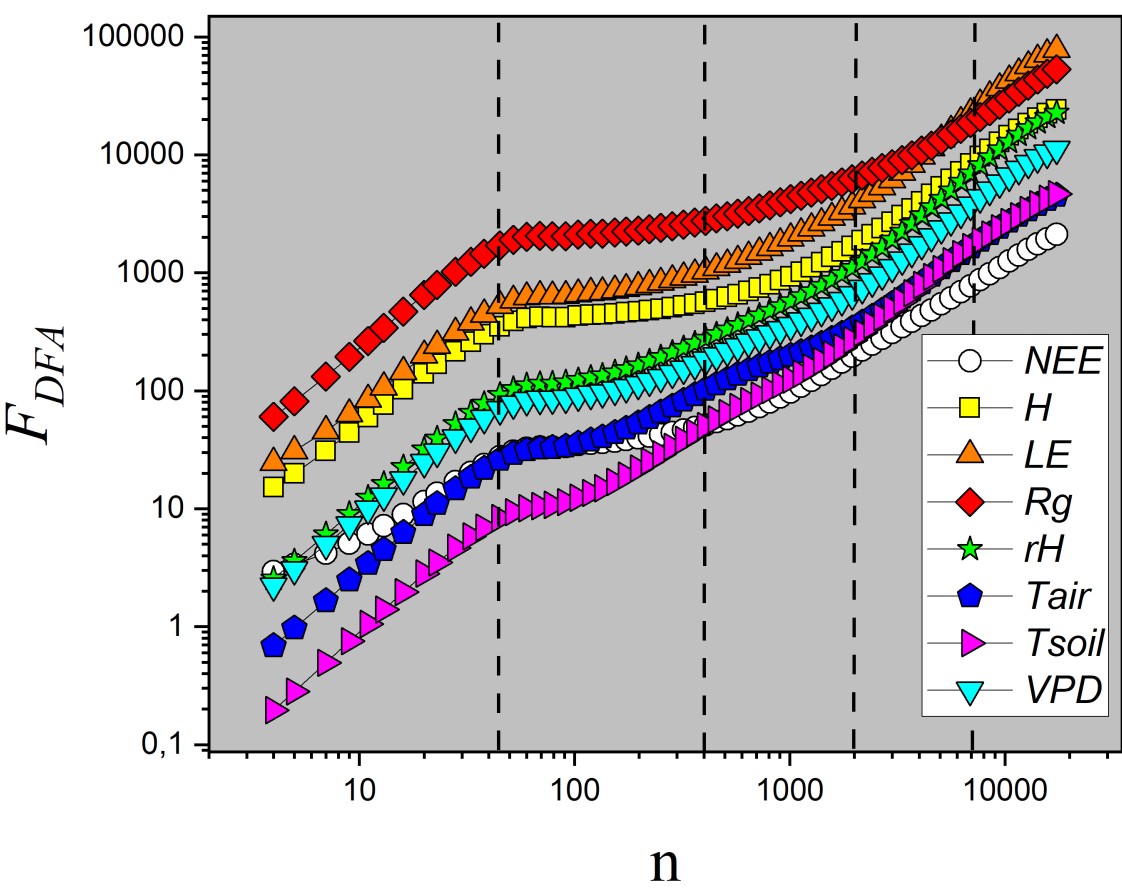

**Figure A6.** Plot of $F_{DFA}$ as a function of the temporal scale $n$ for the eight variables. In this analysis, $n$ represents the number of observation windows, each with a 30-minute duration. The vertical lines indicate the time scales of interest: daily ($n = 48$), weekly ($n = 336$), monthly ($n = 1440$), and quarterly ($n = 4320$).



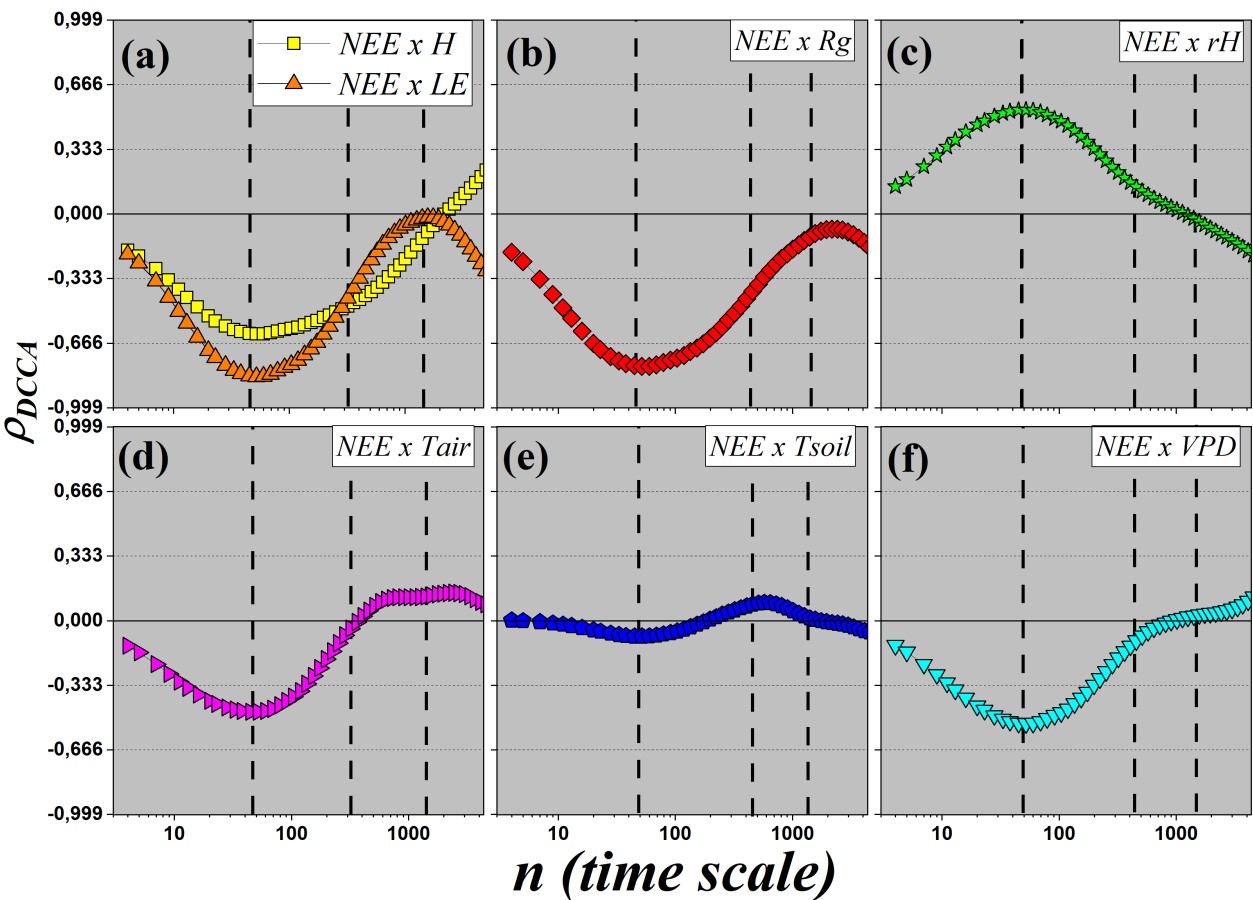

**Figure A7.** DCCA cross-correlation coefficient ($\rho_{\text{DCCA}}$) as a function of the time scale $n$. Each panel displays the cross-correlation between the carbon flux ($NEE$) and one of the following micro-meteorological variables: (a) sensible ($H$) and latent ($LE$) heat fluxes; (b) global radiation ($Rg$); (c) relative air humidity ($rH$); (d) air temperature ($Tair$); (e) soil temperature ($Tsoil$); and (f) vapor pressure deficit ($VPD$).

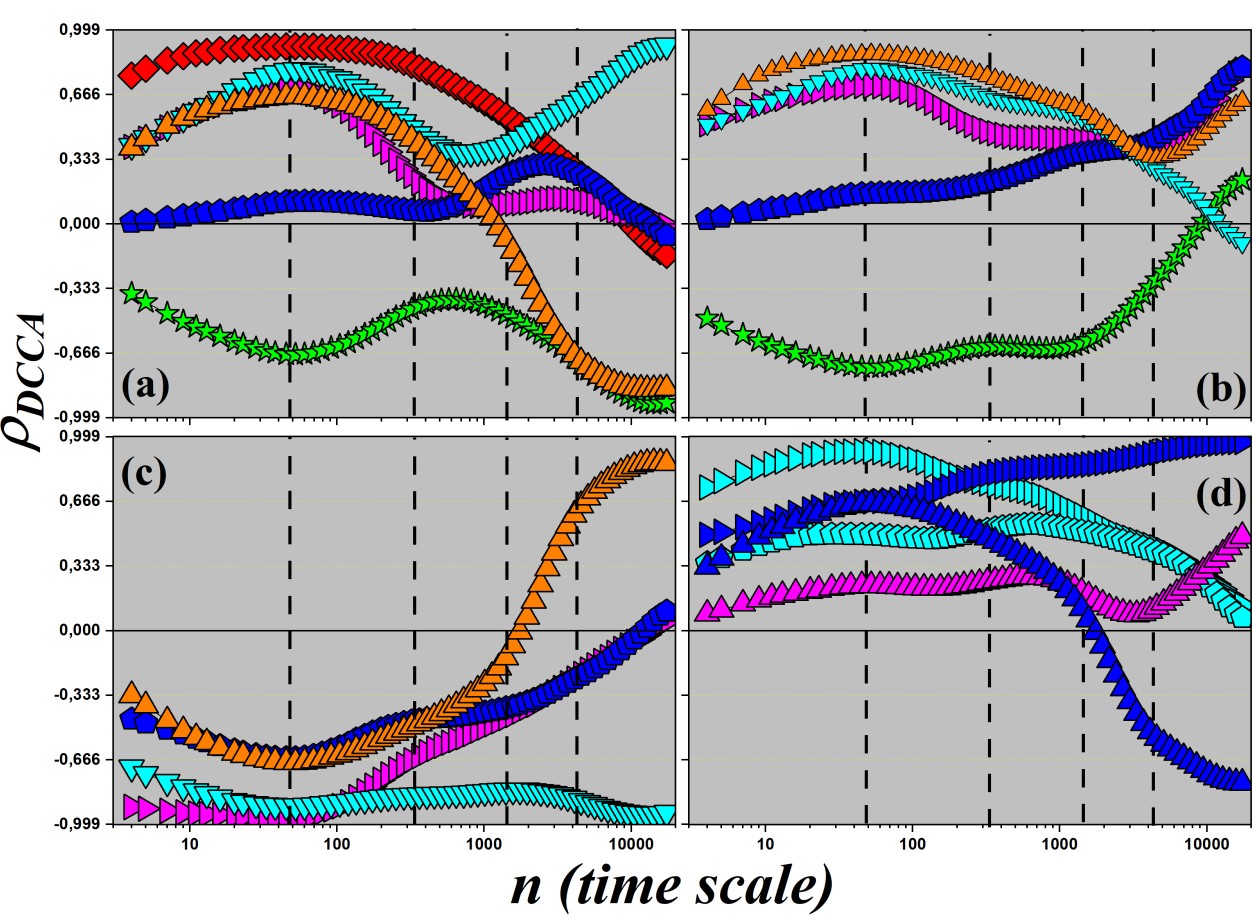

**Figure A8.** Detrended cross-correlation coefficient, $\rho_{\mathrm{DCCA}}$, as a function of time scale $n$ for the variable pairs in the study except $NEE$.



**Appendix B: Tables**

**Table A1.** Interpretation of the DFA Exponent $\alpha$.

| Range of $\alpha$ | Interpretation |
| --- | --- |
| $\alpha = 0.5$ | White Noise: No correlation, random values. |
| $0 < \alpha < 0.5$ | Anti-persistence: High values are followed by low values and vice versa. |
| $0.5 < \alpha < 1$ | Persistence: High values are followed by high values and vice versa. |
| $\alpha \approx 1$ | 1/f Noise: Long-range correlations. |
| $\alpha > 1$ | Non-stationary: Trend present, variance increases over time. |





**Table A2.** Interpretation of $\rho_{\mathrm{DCCA}}$ Coefficient Values

| Value of $\rho_{\mathrm{DCCA}}$ | Interpretation |
| --- | --- |
| -1.000 | perfect anti cross-correlation |
| (-1.000; -0.666] | strong anti cross-correlation |
| (-0.666; -0.333] | moderate anti cross-correlation |
| (-0.333; 0.000) | weak anti cross-correlation |
| 0.000 | no cross-correlation |
| (0.000; 0.333] | weak cross-correlation |
| (0.333; 0.666] | moderate cross-correlation |
| (0.666; 1.000) | strong cross-correlation |
| 1.000 | perfect cross-correlation |





**Table A3.** Descriptive statistics of the eight time series, with $N = 75386$ observations.

|          | $NEE$ (umolm$^{-2}$s$^{-1}$) | $H$ (Wm$^{-2}$) | $LE$ (Wm$^{-2}$) | $Rg$ (Wm$^{-2}$) | $rH$ (%) | $T_{air}$ °C | $T_{soil}$ °C | $VPD$ (hPa) |
|----------|------|------|------|------|------|------|------|------|
| Mean     | 0.54 | 36.0 | 78.5 | 224 | 73.8 | 26.4 | 29.8 | 12.6 |
| Median   | 1.44 | 0.9 | 31.6 | 7.3 | 79.2 | 26.0 | 30.1 | 7.7 |
| Mode     | -10.10 | 101.0 | 108.0 | 0.00 | 93.7 | 24.1 | 29.7 | 0.00 |
| $sd$     | 6.79 | 66.8 | 105.0 | 308 | 20.0 | 5.43 | 3.00 | 13.9 |
| Minimum  | -35.00 | -188.0 | -244.0 | 0.00 | 11.8 | 1.3 | 18.2 | 0.00 |
| Maximum  | 35.00 | 602.0 | 699.0 | 1270 | 99.8 | 42.6 | 39.6 | 83.5 |
| Skewness | -0.17 | 1.7 | 1.8 | 1.09 | -0.80 | -0.14 | -0.46 | 1.48 |
| Kurtosis | 3.50 | 2.5 | 2.8 | -0.235 | -0.293 | 0.190 | 0.296 | 1.99 |





**Table A4.** DFA Exponent

| Index | Daily | Weekly | | Monthly | Quarterly |
|-------|-------|--------|------|---------|-----------|
| $NEE$ | 0.96 | 0.24 | | 0.85 | 1.07 |
| $H$ | 1.31 | 0.19 | | 0.67 | 1.23 |
| $Rg$ | 1.40 | 0.17 | | 0.52 | 0.81 |
| $LE$ | 1.33 | 0.25 | | 0.80 | 1.39 |
| $rH$ | 1.48 | 0.46 | | 0.85 | 1.43 |
| $T_{air}$ | 1.51 | 0.32 | 0.79 | 0.71 | 1.23 |
| $T_{soil}$ | 1.55 | 0.42 | 1.05 | 1.02 | 1.45 |
| $VPD$ | 1.43 | 0.40 | | 0.78 | 1.36 |



**Appendix B: Results of $\rho_{\text{DCCA}}$ for the other pairs of variables.**

Figure B1 presents the results of the detrended cross-correlation coefficients $\rho_{\text{DCCA}}$ calculated for all pairs of variables, except for $NEE$. This analysis was conducted with the goal of providing an overview of the degree of interdependence among
the microclimatic variables. The decision to initially exclude $NEE$ allows for a comparative view of the internal correlations within the physical system, reserving the analysis of correlations between these variables and the ecosystem carbon flux ($NEE$) for a later and more detailed stage.

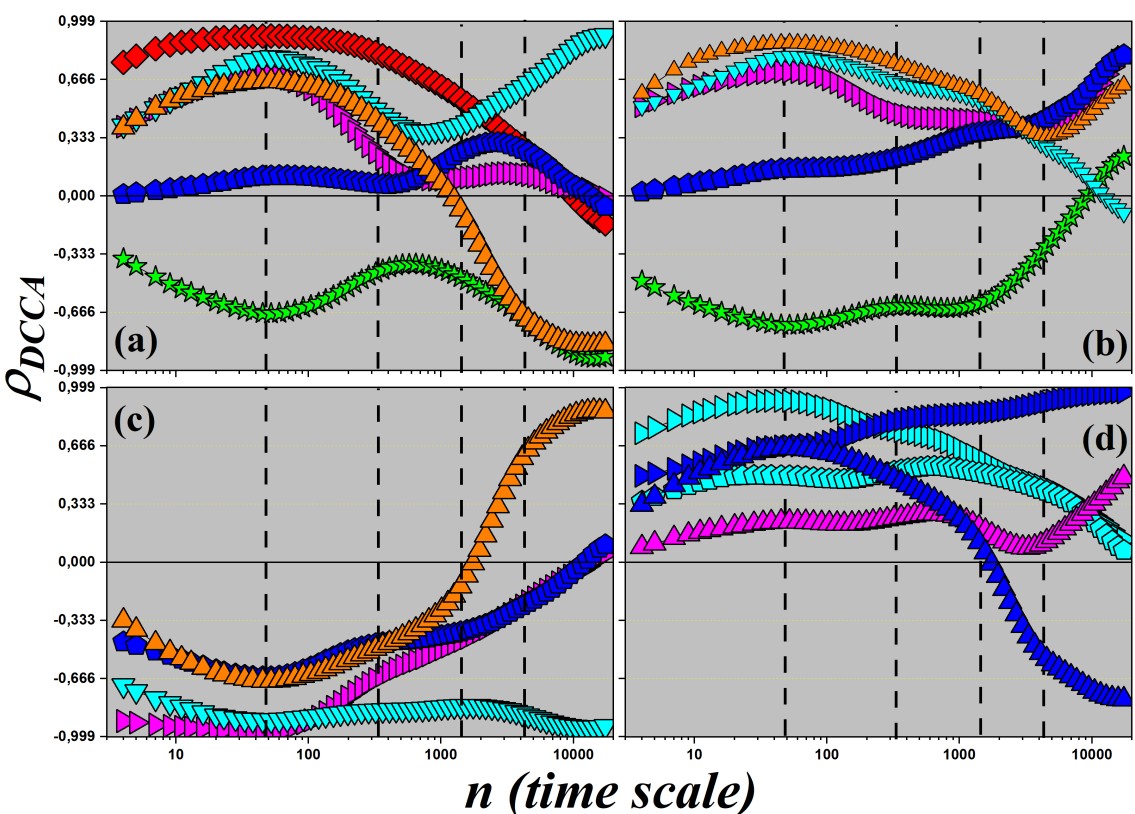

**Figure B1.** Detrended cross-correlation coefficient, $\rho_{\text{DCCA}}$, as a function of time scale $n$ for all variable pairs in the study, except $NEE$.

    In Fig. B1(a), the $\rho_{\text{DCCA}}$ values are shown for the correlations between sensible heat flux ($H$) and the other variables. At small scales, a moderate correlation is observed between $H$ and most variables, with particular emphasis on $Rg$, which exhibits
strong correlation from the smallest scales up to the weekly scale. Beyond this point, the correlation with $Rg$ progressively decreases, becoming moderate at the monthly scale and weak at the quarterly scale, eventually reaching values close to zero or even weak anti-correlation at larger scales. The correlation between $H$ and relative humidity ($rH$) is consistently negative across all scales, with strong anti-correlation at the daily scale, suggesting that high $rH$ values are associated with lower




energy availability for heating the air and soil, favoring instead evapotranspiration processes ($LE$). At the weekly, monthly,
and quarterly scales, the anti-correlation between $H$ and $rH$ remains moderate. Soil temperature ($T_{soil}$) shows very weak or
nonexistent correlation with $H$ at small, daily, and weekly scales. At the quarterly scale, a weak correlation emerges, which
disappears again at larger scales. At broader time scales, strong correlation is observed between $H$ and $VPD$, as well as strong
anti-correlation with $LE$ and $rH$, while the other variables show no significant correlation with $H$.

In Fig. B1(b), the correlations between global radiation ($Rg$) and the other variables are shown. At small scales, $Rg$ presents
moderate correlation with $LE$, $T_{air}$, and $VPD$, which intensifies to strong correlation at the daily scale. As the scale increases,
the $\rho_{\mathrm{DCCA}}$ values for these variables decrease: $LE$ maintains strong correlation at the daily scale and moderate at the monthly
and quarterly scales; $T_{air}$ exhibits a similar pattern, with a drop at the monthly scale and a return to strong correlation at larger
scales; $VPD$, in turn, shows decreasing correlation until reaching weak anti-correlation at longer scales. Soil temperature
($T_{soil}$) initially displays very weak correlation with $Rg$ at small scales, but shows continuous growth: weak correlation at
daily and monthly scales, moderate at the quarterly scale, and strong at larger scales. The correlation between $Rg$ and $rH$ is
negative at small scales, indicating moderate anti-correlation. This anti-correlation intensifies at the daily scale, decreases to
moderate between the weekly and monthly scales, and then gradually reverses sign, transitioning from weak anti-correlation at
the quarterly scale to weak (positive) correlation at larger scales.

In Fig. B1(c), correlations are presented between relative humidity ($rH$) and the variables $LE$, $T_{air}$, $T_{soil}$, and $VPD$.
At small scales, all correlations are negative. $LE$ and $T_{soil}$ show moderate anti-correlation, while $T_{air}$ and $VPD$ exhibit
strong anti-correlation. The anti-correlation between $rH$ and $T_{air}$ is particularly notable at the daily scale, with $\rho_{\mathrm{DCCA}}$ values
near -1. As the scale increases, this relationship weakens: moderate anti-correlation at the weekly and monthly scales, weak
anti-correlation at the quarterly scale, and no correlation at larger scales. The behavior of $T_{soil}$ is similar: it shows moderate
anti-correlation up to the monthly scale, weak anti-correlation at the quarterly scale, and no correlation at broader scales. The
correlation between $rH$ and $VPD$ is negative across the entire series, reflecting the expected inverse relationship between
relative humidity and vapor pressure deficit — reaching values close to -1 at larger scales. Meanwhile, the correlation between
$rH$ and $LE$ starts as moderate anti-correlation at small scales, becoming strong at the daily scale and weak at the weekly scale.
From the monthly scale onward, the correlation increases sharply and becomes positive: moderate at the quarterly scale and
strong at larger scales, reflecting the role of humidity in sustaining evapotranspiration during prolonged periods.
In Fig. B1(d), the correlations between the remaining variable pairs are shown. The two cyan lines represent the correlations
between $VPD$ and $T_{air}$, and between $VPD$ and $T_{soil}$. The blue curves represent the correlations between $T_{soil}$ and $LE$, and
between $T_{soil}$ and $T_{air}$. The magenta curve refers to the correlation between $LE$ and $T_{air}$. In general, all these variables show
positive correlations from small scales to the monthly scale. From the monthly scale onward, the correlation between $LE$ and
$T_{soil}$ rapidly decreases, reaching strong anti-correlation at larger scales. The following points are noteworthy:
- At the daily scale, the correlation between $T_{air}$ and $VPD$ is very strong, indicating that high temperatures are strongly
associated with increased water stress;
- The correlation between $LE$ and $T_{air}$ is weak at small scales and also at the quarterly scale, showing moderate correlation
only at larger scales;



- The correlation between $T_{air}$ and $T_{soil}$ displays an approximately linear pattern, moving from moderate correlation at small
scales to very strong correlation at the largest scales.

*Author contributions.* T.S., A.A., P.A., and G.Z. planned the research; P.A. performed the measurements and processed the data; G.Z. devised the methodology; T.S., A.A., and G.Z. analyzed the data; T.S. wrote the manuscript draft; A.A. and G.Z. reviewed and edited the manuscript.

*Code and data availability.* All raw data and code are available from the corresponding authors upon reasonable request

*Competing interests.* The authors declare that they have no conflict of interest.

*Acknowledgements.* The authors acknowledge the financial support from the Brazilian agencies CNPq and CAPES. Special thanks to the Graduate Program in Physics (Master's and PhD) at the Federal University of Mato Grosso (UFMT) for the opportunity to pursue this research, and to the State University of Mato Grosso (UNEMAT) and Federal University of Mato Grosso (UFMT) for its continued encouragement of academic qualification and scientific development.

Zebende acknowledges financial support from CNPq Grant 302867/2025-2.

The Article Processing Charge for the publication of this research was funded by the Coordenação de Aperfeiçoamento de Pessoal de Nível Superior - CAPES (ROR identifier: 00x0ma614). For open access purposes, the authors applied the Creative Commons CC BY license to any accepted version of the article.



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
