# Peer review of "Multi-scale dynamics of carbon dioxide flux and its environmental drivers in the Pantanal wetland"

_EGUsphere, 2025_

## Author Response (AR1)

**Final response to Editor**

**Multi-scale dynamics of carbon dioxide flux and its environmental drivers in the Pantanal wetland**
**Tarcis Santos, et al**

We sincerely thank both reviewers for their thoughtful and constructive comments, which have significantly improved the clarity and rigor of our manuscript. Below, we provide a point-by-point response to each comment and a summary of the changes implemented.

**Response to Reviewer 1**

We thank Reviewer 1 for the thorough and highly positive feedback on the manuscript's methodological robustness, clarity, and contribution. We are pleased that the application of Detrended Cross-Correlation Analysis (DCCA) was considered appropriate and effective, and that the conclusions regarding the scale-dependence of carbon-climate interactions were well received.

**R1.1 Comment on Redundancy (Line 370):**

- **Request:** Revisit this section as these details have already been provided in lines 357 and 356.
- **Action Taken:** The text in the Results and Discussion section (originally around line 370, concerning the DFA results summary) was revised to remove redundancy and ensure a more concise and fluid presentation of information, avoiding unnecessary repetition of results already described earlier in the section.

**R1.2 Comment on Figures (Map and Site Picture):**

- **Request:** It would be great to include a map of the site, and a picture of the site set up as an insert – preferably.
- **Action Taken:**
  (a) A geographical location map of the study area, precisely indicating the measurement tower location (Fazenda Nossa Senhora do Carmo), was added to the manuscript as **Figure 1** (Section 2.1).
  (b) **Section 2.1 (Materials)** was updated with a photograph of the micrometeorological tower and sensor setup as an inset in the same figure.

**Response to Reviewer 2**

We thank Reviewer 2 for the in-depth evaluation of the methodology and for the suggestions for specific improvements in the interpretation of coefficients. The Reviewer is correct in pointing out the need for greater precision in the classification of the $\alpha_{DFA}$ exponents and in the table captions.

**R2.1 Comment on $\alpha_{DFA} > 1.5$ (Table 1 and 3):**

- **Request:** Wouldn't it be interesting to add the $\alpha_{DFA}$ value for values greater than 1.5 to Table 1 and justify this in the text? (Observation about $T_{air}$ and $T_{soil}$ at $\alpha \approx 1.51/1.55$).
- **Action Taken:**
  (a) The **Table for Interpreting $\alpha_{DFA}$ Exponents** was significantly expanded to include more detailed classifications for the non-stationary regime ($\alpha > 1$), as suggested. The new categories are:
    - $1 < \alpha < 1.5$: Persistent, non-stationary (fractional Brownian motion).
    - $\alpha \approx 1.5$: Brownian noise (random walk; integrated white noise).

$-\ \alpha > 1.5$: Very strong trend / super... (super-diffusive).

    (b) The text in **Section 2.2.1 (DFA)** was supplemented with an explicit justification for the exponent $\alpha \approx 1.5$ (Brownian Noise), differentiating it from generic non-stationarity ($\alpha > 1$), to provide a more solid theoretical basis for interpreting the $T_{air}$ and $T_{soil}$ results.

**R2.2 Comment on Table 2 Caption ($\rho_{DCCA}$ Interpretation):**

- **Request:** In Table 2, the authors present the interpretation of the $\rho_{DCCA}$ coefficient, but describe it in the caption as the DFA exponent. I believe this should be revised.

- **Action Taken:** The caption for the Table of $\rho_{DCCA}$ Coefficient Interpretation (original Table 2, now Table A2 in the Appendix) was revised and corrected to accurately reflect the table's content, mentioning the $\rho_{DCCA}$ coefficient.

**R2.3 Comment on Table 3 Caption (Calculated $\alpha_{DFA}$ Values):**

- **Request:** The same mistake occurs in Table 3. The table shows the calculated $\alpha_{DFA}$ values, but the caption describes it as the interpretation of $\rho_{DCCA}$. I also suggest reviewing this.

- **Action Taken:** The caption for the Table presenting the calculated $\alpha_{DFA}$ exponent values (original Table 3, now Table 3 in the main body) was revised and corrected to precisely describe the content: the $\alpha_{DFA}$ exponent values at different time scales.

**R2.4 Comment on Persistence Levels ($\alpha_{DFA}$):**

- **Request:** I did not identify a clear description of the persistence levels in the text; only the information that values above 0.5 indicate persistent behavior. The text mentions 0.85 as moderate persistence, but to validate this statement, the authors should explain what they consider to be weak, moderate, and strong persistence. Personally, I believe it would be sufficient to indicate only the term persistence, without the subdivision.

- **Action Taken:** The text in the Results and Discussion section was revised to **remove the subjective subdivision** of the $\alpha_{DFA}$ exponents (e.g., "weak persistence", "moderate persistence", "strong persistence"). Following the reviewer's suggestion, the manuscript now primarily uses the more generic and formal term **"Persistence"** for values $0.5 < \alpha_{DFA} < 1.0$, ensuring consistency with the Interpretation Table and standard DFA literature. Subdivisions (weak, moderate, strong) are maintained only for the $\rho_{DCCA}$ cross-correlation coefficient, where they are widely validated in specific literature.

**List of Relevant Changes Made in the Manuscript**

The following changes were made to the manuscript (new version: `template_tarcis.pdf`) relative to the old version (`egusphere-2025-4102-manuscript-version2.pdf`):

**Added/Requested Changes (Reviewers)**

1. **Inclusion of Location Figure:** Addition of a figure (**Figure 1**) with the geographical location map and the exact point of the measurement tower, as requested by Reviewer 1.

2. **Revision of $\alpha_{DFA}$ Table:** The Table for Interpreting $\alpha_{DFA}$ Exponents (formerly Table 1) was expanded (now **Table A1** in the Appendix) to include detailed classifications of non-stationary regimes ($1 < \alpha_{DFA} < 1.5$, $\alpha_{DFA} \approx 1.5$, and $\alpha_{DFA} > 1.5$), addressing Reviewer 2.

3. **DFA Justification in Text:** Addition of text in **Section 2.2.1 (DFA)** to justify and differentiate the case of Brownian Noise ($\alpha_{DFA} \approx 1.5$) from general non-stationarity ($\alpha_{DFA} > 1$), as suggested by Reviewer 2.

4. **Correction of Captions:** The captions of the tables presenting the interpretation of $\rho_{DCCA}$ (formerly Table 2) and the calculated values of $\alpha_{DFA}$ (formerly Table 3) were corrected to avoid incorrect coefficient description, addressing Reviewer 2.

5. **Removal of Redundancy:** The text in the **Results and Discussion Section** was revised to remove the redundancy identified by Reviewer 1 (around the old Line 370).

6. **Consistency in Persistence:** The text was revised to remove or clarify subjective subdivisions of $\alpha_{DFA}$ (e.g., "moderate persistence"), consistently using the term "Persistence" for $0.5 < \alpha_{DFA} < 1.0$, as suggested by Reviewer 2.

**Structural and Other Changes**

7. **Table Reorganization:** The tables were revised and reorganized. All of them are present in the main body of the text and in Appendix B.

8. **Figure Numbering:** The figure numbering was adjusted (e.g., the old Figure 1, now **Figure 2**, is the time series data).